


**Ideas and perspectives: A strategic assessment of methane and nitrous oxide measurements**
**in the marine environment**
Samuel T. Wilson[1], Alia N. Al-Haj[2], Annie Bourbonnais[3], Claudia Frey[4], Robinson W.
Fulweiler[2,5], John D. Kessler[6], Hannah K. Marchant[7], Jana Milucka[7], Nicholas E. Ray[5], Parv
Suntharalingham[8], Brett F. Thornton[9], Robert C. Upstill-Goddard[10], Thomas S. Weber[6], Damian
L. Arévalo-Martínez[11], Hermann W. Bange[11], Heather M. Benway[12], Daniele Bianchi[13], Alberto
V. Borges[14], Bonnie X. Chang[15], Patrick M. Crill[9], Daniela A. del Valle[16], Laura Farías[17],
Samantha B. Joye[18], Annette Kock[11], Jabrane Labidi[19], Cara C. Manning[20#], John W. Pohlman[21],
Gregor Rehder[22], Katy J. Sparrow[23], Philippe D. Tortell[20], Tina Treude[13,19], David L. Valentine[24],
Bess B. Ward[25], Simon Yang[13], Leonid N. Yurganov[26]
[1]University of Hawai'i at Manoa, Daniel K. Inouye Center for Microbial Oceanography: Research and
Education (C-MORE), Hawai'i, USA
[2]Boston University, Department of Earth and Environment, Massachusetts, USA
[3]University of South Carolina, School of the Earth, Ocean and Environment, South Carolina, USA
[4]University of Basel, Department of Environmental Science, Basel, Switzerland
[5]Boston University, Department of Biology, Massachusetts, USA
[6]University of Rochester, Department of Earth and Environmental Science, New York, USA.
[7]Max Planck Institute for Marine Microbiology, Department of Biogeochemistry, Bremen,
Germany
[8]University of East Anglia, School of Environmental Sciences, Norwich, UK.
[9]Stockholm University, Department of Geological Sciences and Bolin Centre for Climate Research,
Stockholm, Sweden
[10]Newcastle University, School of Natural and Environmental Sciences, Newcastle upon Tyne, UK
[11]GEOMAR Helmholtz Centre for Ocean Research Kiel, Düsternbrooker Weg 20, 24105 Kiel, Germany
[12]Woods Hole Oceanographic Institution, Marine Chemistry and Geochemistry, Massachusetts, USA,
[13]University of California Los Angeles, Department of Atmospheric and Oceanic Sciences, California,
USA
[14]University of Liège, Chemical Oceanography Unit, Liège, Belgium
[15]University of Washington, Joint Institute for the Study of the Atmosphere and Ocean, Seattle, USA
[16]University of Southern Mississippi, Division of Marine Science, Mississippi, USA





[17]University of Concepción, Department of Oceanography and Center for Climate Research and
Resilience (CR2), Concepción, Chile
[18]University of Georgia, Department of Marine Sciences, Georgia, USA
[19]University of California, Los Angeles, Department of Earth, Planetary, and Space Sciences, Los
Angeles, California, USA
[20]University of British Columbia, Department of Earth, Ocean and Atmospheric Sciences, British
Columbia, Vancouver, Canada
[21]U.S. Geological Survey, Woods Hole Coastal and Marine Science Center, Woods Hole, USA
[22]Leibniz Institute for Baltic Sea Research Warnemünde, Rostock, Germany
[22]Florida State University, Department of Earth, Ocean, and Atmospheric Science, Florida, USA
[24]University of California Santa Barbara, Department of Earth Science, California, USA
[25]Princeton University, Geoscience Department, New Jersey, USA
[26]University of Maryland Baltimore County, Baltimore, USA
[#]Current address: Plymouth Marine Laboratory, Plymouth, UK

**Abstract**. In the current era of rapid climate change, accurate characterization of climate-
relevant gas dynamics - namely production, consumption and net emissions - is required for all
biomes, especially those ecosystems most susceptible to the impact of change.  Marine
environments include regions that act as net sources or sinks for a number of climate-active trace
gases including methane ($CH_4$) and nitrous oxide ($N_2O$).  The temporal and spatial distributions
of $CH_4$ and $N_2O$ are controlled by the interaction of complex biogeochemical and physical
processes.  To evaluate and quantify the importance of these mechanisms relevant to marine $CH_4$
and $N_2O$ cycling requires a combination of traditional scientific disciplines including
oceanography, microbiology, and numerical modeling.  Fundamental to all of these efforts is
ensuring that the datasets produced by independent scientists around the world are comparable
and interoperable.  Equally critical is transparent communication within the research community
about the technical improvements required to increase our collective understanding of marine
$CH_4$ and $N_2O$.  An Ocean Carbon & Biogeochemistry (OCB) sponsored workshop was organized
to enhance dialogue and collaborations pertaining to marine $CH_4$ and $N_2O$.  Here, we summarize
the outcomes from the workshop to describe the challenges and opportunities for near-future
$CH_4$ and $N_2O$ research in the marine environment.



## 1. Background

The most abundant greenhouse gases in the troposphere, excluding water vapor, are carbon dioxide ($CO_2$), methane ($CH_4$), and nitrous oxide ($N_2O$). Together they account for more than 80% of the total radiative forcing (IPCC, 2013) and their current tropospheric mole fractions and rates of increase are unprecedented in recent Earth history (Ciais et al., 2013; Burke et al., 2020; Fig. 1a). While $CO_2$ is the most abundant of the three greenhouse gases, $CH_4$ and $N_2O$ both have a higher warming potential than $CO_2$ (Montzka et al., 2011). To accurately constrain the contribution of $CH_4$ and $N_2O$ to Earth's radiation budget requires their sources and sinks to be quantified with high resolution at the global scale.

The oceans are a fundamental component of the global climate system and are a net source of tropospheric $CH_4$ and $N_2O$ at the global scale, although local to regional budgets may include both source and sink aspects. There are far fewer marine measurements of dissolved $CH_4$ and $N_2O$ than of dissolved $CO_2$ and while there is substantial international coordination with regard to $CO_2$ analysis, calibration and data reporting, no such coordination yet exists for $CH_4$ and $N_2O$ (Wilson et al. 2018). Given the increasing prominence of climate change on scientific and societal agendas, greater coordination among the marine $CH_4$ and $N_2O$ scientific community to provide more targeted measurements and increase the quality and interoperability of $CH_4$ and $N_2O$ observations is particularly timely.

Despite the lack of an international coordinating framework, there have been important advances in our understanding of marine $CH_4$ and $N_2O$ in numerous research disciplines, ranging from cellular metabolism and model microbial systems to large-scale modeling. For example, recent work identified novel microorganisms and metabolic pathways in the production of $N_2O$ (Trimmer et al., 2016; Caranto and Lancaster, 2017) and $CH_4$ (Repeta et al. 2016; Bižić et al., 2020). Earth system models now incorporate improved $N_2O$ parameterizations to better resolve the ocean's role in the global $N_2O$ cycle (Battaglia and Joos, 2018). New techniques enable the discrimination of ancient and modern dissolved $CH_4$ (Sparrow et al., 2018) and the transfer of $CH_4$-derived carbon to other carbon pools (Pohlman et al., 2011; Garcia-Tigreros and Kessler, 2018). Other technological and analytical advances include improved near-continuous spectroscopic analysis that yield greater sampling resolution in surface waters (e.g. Gülzow et



al., 2011; Arévalo-Martínez et al., 2013; Erler et al., 2015) and the deployment of analytical
devices on robotic vehicles (Nicholson et al., 2018).

These scientific advances and an improvement in the quantity and quality of $CH_4$ and $N_2O$

observations are timely given that large areas of both the open and coastal ocean remain under-
sampled (Fig. 1b). This leads to uncertainty in oceanic $CH_4$ and $N_2O$ inventories, their rates of
production and consumption, and their emissions. This is problematic given that the marine
environment is susceptible to an accelerating rate of anthropogenic change that will continue to
modify the global cycles of carbon and nitrogen into the future. Environmental impacts on
marine $CH_4$ and $N_2O$ distributions include increasing seawater temperatures, decreasing
concentrations of dissolved oxygen ($O_2$), acidification, retreat of ice and mobilization of carbon
substrates from former permafrost, altering coastal run-off, and eutrophication (IPCC, 2019).
These impacts will undoubtedly alter future $CH_4$ and $N_2O$ exchange with the atmosphere, but the
directions and magnitudes of these modified fluxes remains insufficiently understood.

A need to resolve the uncertainties prompted an evaluation of the collective ability of the

international scientific community to accurately determine the distribution and emissions of $CH_4$
and $N_2O$, and the physical-biogeochemical factors that determine them. This became the focus
of a marine $CH_4$ and $N_2O$ workshop hosted by the Ocean Carbon and Biogeochemistry (OCB)
program at Lake Arrowhead, California in October 2018. The workshop considered $CH_4$ and
$N_2O$ equally on the same agenda, even though nearly all field, laboratory, and modeling studies
examine these trace gases separately. The rationale for this dual approach is that $CH_4$ and $N_2O$
share common considerations of the physical, chemical, and microbial processes that dictate their
water-column distributions (Bakker et al., 2014; Bodelier and Steenbergh, 2014). In addition,
many of the analytical procedures for quantifying $CH_4$ and $N_2O$ and the subsequent data quality
assurances share many common requirements. The opportunity to bring a large research
community together to increase dialogue and encourage the cross-fertilization of ideas was thus
considered very valuable. This article articulates the workshop outcomes framed in the context
of current marine $CH_4$ and $N_2O$ research and explores future research opportunities and
challenges.

**2. Coordination of oceanic $CH_4$ and $N_2O$ measurements**



Our understanding of the temporal and spatial distributions of oceanic $CH_4$ and $N_2O$ derives
from over five decades of open ocean and coastal observations, including targeted expeditions,
repeat hydrographic surveys, and time-series monitoring, each of which has been crucial to the
development of our current knowledge (Fig. 2). Targeted programs have enabled invaluable
insights into the role of oxygen deficient zones in $N_2O$ cycling (Babbin et al., 2015; Bourbonnais
et al., 2017; Frey et al., 2020) and the exploration of $CH_4$-rich seeps and vents (Foucher et al.,
2009; Suess, 2010; Boetius and Wenzhöfer, 2013). Basin-scale repeat hydrographic surveys
(e.g. the international GO-SHIP program) have facilitated extensive water-column mapping to
identify relevant water masses and evaluate ventilation rates (e.g. de la Paz et al., 2017). Other
oceanic surveys have focused exclusively on surface sampling, using continuous equilibrator
systems connected to various gas analyzers to yield high-resolution surface concentration fields
of $CH_4$ and $N_2O$ (Gülzow et al., 2013; Erler et al., 2015; Kodovska et al., 2016; Thornton et al.,
2016a; Pohlman et al., 2017). In contrast, sustained long-term time-series measurements of $CH_4$
and $N_2O$ at fixed monitoring stations are relatively few, but they span a range of latitudes and
biogeochemical provinces (Fig. 2). The time-series observations provide the contextual
background for seasonal and interannual variation that allow long-term temporal trends and
episodic events to be identified and evaluated (Farías et al., 2015; Wilson et al., 2017; Ma et al.,
2019). Overall, the majority of measurements enable the variability in marine $CH_4$ and $N_2O$ to
be quantified at the mesoscale or greater (i.e. from hundreds of kilometers to ocean basins), with
monthly to annual resolution but there are substantially fewer datasets at the sub-mesoscale level
(i.e. <10 km and hours to days) (Fig. 3). A major reason for the limited sampling at the sub-
mesoscale level is that it necessitates high-resolution measurements to resolve the heterogeneous
variability that exists at these time-space scales. Such analyses have only recently become
technically feasible (discussed in more detail in Section 6).
Until recently there has been no formal coordination of observations across the $CH_4$ and $N_2O$
scientific community. In response to this, a Scientific Committee on Oceanic Research (SCOR)
Working Group was initiated in 2014 entitled: '*Dissolved $N_2O$ and $CH_4$: Working towards a*
*global network of ocean time series measurements*'. A major goal of the SCOR Working Group
was to unite the international community in joint activities conceived to improve and inform
seagoing $CH_4$ and $N_2O$ analyses. An important activity was the preparation and distribution of
common, combined gaseous $CH_4$ and $N_2O$ standards to twelve international laboratories, with





the aim of improving and standardizing calibration (Bullister et al., 2017). A subsequent inter-
comparison of discrete seawater samples included the use of these standards and revealed the
variability between laboratories. While there were some encouraging results from the
intercomparison, such as the agreement between individual laboratories using contrasting
techniques, overall a large range was observed in $CH_4$ and $N_2O$ concentration data generated by
the participating laboratories (Wilson et al., 2018). Such analytical discrepancies weaken our
collective ability as a community to evaluate temporal-spatial variability in marine $CH_4$ and $N_2O$.
The discrepancies also highlighted the need for Standard Operating Protocols (SOPs) for $CH_4$
and $N_2O$ analyses to facilitate standardization of sampling, measurement, and calibration, as well
as the reporting of data and accompanying metadata in common repositories. The SOPs are
currently in preparation with intended publication on the Ocean Best Practices network.
A data repository for oceanic $CH_4$ and $N_2O$ data known as the MarinE MEthane and NiTrous
Oxide database (MEMENTO) was established in 2009 (Bange et al., 2009; Kock and Bange,
2015). MEMENTO is now sufficiently mature to support descriptions of the broad-scale surface
distributions of $CH_4$ and $N_2O$ (e.g. Suntharalingam et al., 2012; Zamora and Oschlies, 2014;
Buitenhuis et al., 2018; Battaglia and Joos, 2018). Machine-learning mapping recently identified
$CH_4$ and $N_2O$ distributions and various physical and biogeochemical predictor variables (e.g.
depth, temperature, salinity, $O_2$, nutrients, primary production) (Weber et al., 2019; Yang et al.,
2020, Fig. 4). The application of gas transfer algorithms to the extrapolated oceanic $CH_4$ and
$N_2O$ distributions helped decrease the uncertainty in estimates of global air-sea exchange fluxes
(Fig. 4c), thereby fulfilling one of the key goals of MEMENTO (Bange et al., 2009). Net global
open ocean emissions of $N_2O$ are now similarly estimated at 3–5 Tg N $yr^{-1}$ by both Yang et al.
(2020) and the Global Nitrous Oxide Project (Tian et al., 2020). In comparison, net global ocean
$CH_4$ emissions from machine-learning mapping were estimated at 6–12 Tg $CH_4$ $yr^{-1}$ (Weber et
al., 2019), compared to 9–22 Tg $CH_4$ $yr^{-1}$ in the most up-to-date $CH_4$ synthesis (Saunois et al.,
2020). However, the narrower range for machine-learning derived $CH_4$ emissions retains high
uncertainty in regions such as the Arctic, where emissions are highly heterogeneous and
compounded by seasonal ice cover. Identifying the causes for uncertainty in high emission
regions will greatly aid future sampling campaigns, as is discussed in the following sections.

**3. Methane in marine environments**



In the surface waters of tropical and temperate oceans, the low supersaturation of $CH_4$ is
driven by aerobic production arising from the decomposition of methyl phosphonate in
phosphorus-depleted waters (Karl et al. 2008, Sosa et al., 2020), the degradation of methylated
sulfur compounds by phytoplankton (Klintzsch et al., 2019), and other processes (Schmale et al.,
2018). Deep within the ocean's pelagic interior, $CH_4$ is weakly undersaturated reflecting
depletion via microbial oxidation (Reeburgh 2007; Weber et al., 2019). Towards the coastline,
$CH_4$ supersaturation increases by orders of magnitude (Figure 5b), reflecting terrestrial inputs
(e.g. river and groundwater) and $CH_4$ diffusion and ebullition from shallow anoxic methane rich
sediments (Zhang et al., 2008; Borges et al., 2016; Upstill-Goddard and Barnes, 2016).
Supersaturation of $CH_4$ occurs frequently in the Arctic Ocean and its relatively shallow marginal
seas with the most extreme values observed in the Eurasian Arctic (e.g. Shakhova et al., 2010;
Damm et al., 2015; Kosmach et al., 2015; Thornton et al, 2016a; Fenwick et al., 2017).
Terrestrial and subsea permafrost are potential $CH_4$ sources to shelf waters in addition to $CH_4$
hydrates that are found in marginal shelves globally (Ruppel and Kessler, 2017). Large point
source $CH_4$ emissions, such as seafloor gas seeps can be large sources to the atmosphere in small
localized areas (e.g. Thornton et al., 2020), but these sites remain particularly difficult to
parameterize in models. This reflects limited observations and a poor understanding of their
spatial distributions, the driving mechanisms, and the wider context within the carbon cycle. For
example, the upwelling of cold, nutrient-rich water that accompanies $CH_4$ ascending the water
column stimulates $CO_2$ consumption by photosynthesizing phytoplankton, rendering such $CH_4$
seeps an overall net sink for climate-forcing gases (Pohlman et al., 2017). Recent work using
thermal infrared satellite retrievals indicates increased high-latitude oceanic $CH_4$ release in late
autumn, coincident with pycnocline breakdown and a deepening of the ocean mixed layer depth
thereby bringing deep $CH_4$ to the surface (Yurganov et al., 2019). This is especially notable in
the Kara and Barents Seas, but the remote observations have not yet been confirmed by surface
ocean measurements which are difficult and therefore rare, except during the Arctic summer.
Seabed $CH_4$ emissions are hypothesized to increase in a warming ocean through the
decomposition of gas hydrates, the degradation of subsea permafrost under some high-latitude
seas, and the increased biodegradation of sediment carbon (Romanovskii et al., 2005; Biastoch et
al., 2011; Ruppel and Kessler, 2017; Borges et al., 2019). Effort is thus focused on quantifying
the fraction of $CH_4$ generated in or released from marine sediments that ultimately enters the



atmosphere, particularly on shallow continental shelves and in coastal ecosystems.  Natural
stable isotopes have been used to inform spatial and temporal changes in dissolved $CH_4$
concentrations (e.g. Pack et al., 2011; Mau et al., 2012; Weinstein et al., 2016; Leonte et al.,
2017; Chan et al., 2019) and incubation experiments with added stable isotopes and radiotracers
have helped elucidate how oxidation (anaerobically in sediments and aerobically in the water
column), ebullition (where $CH_4$ pore water partial pressure exceeds sediment hydrostatic
pressure), and subsequent bubble dissolution in the water column interact to mitigate $CH_4$
emissions to air (Steinle et al., 2015; Jordan et al., 2020).  The information deriving from these
various approaches is inherently different but complementary.  Isotope tracer incubations provide
snapshots of rates specific to the methanotrophic community and $CH_4$ concentration at the time
of sampling, whereas concentrations and isotopic gradients are used to infer *in situ* rates
integrated over space and time.  A recent study deployed a remotely operated vehicle to examine
the isotopic fractionation of $CH_4$ during bubble ascent and used this to constrain the extent of
bubble dissolution (Leonte et al., 2018).  This work demonstrated an experimental approach
established for broadly constraining water column $CH_4$ cycling directly from a surface research
vessel.

Despite the range of analytical and experimental approaches available, determining whether

the origin of the emitted $CH_4$ is seafloor release or aerobic production in the upper water column
remains problematic.  To date there is no straightforward way to routinely distinguish between
seafloor derived and water column generated $CH_4$ for all locations.  Even so, stable carbon and
hydrogen isotope measurements (i.e. $\delta^{13}C\text{-}CH_4$ and $\delta^2H\text{-}CH_4$) combined with ancillary data may
provide valuable source information.  For example, combining these measurements with the ratio
of $CH_4$ to higher order hydrocarbons (e.g. ethene ($C_2H_4$) and ethane ($C_2H_6$)) can be used to infer
for example, whether the origin of the $CH_4$ is thermogenic, sub-seafloor, or biogenic within the
water column (Whiticar, 1999; Pohlman et al., 2009; Lan et al., 2019).  Continuous shipboard
measurement of $CH_4$ isotopes in surface water (e.g. Pohlman et al., 2017) and in the atmospheric
boundary layer (Pankratova  et al., 2019; Berchet et al., 2020) are now possible and they have
been used in combination with atmospheric inversion models to characterize and discriminate
marine-emitted $CH_4$ from other sources (Berchet et al., 2020).  Application of this method to
land-based monitoring stations appears promising for apportioning $CH_4$ emissions from various
marine regions and sources (Thonat et al., 2019).  Additionally, in regions where aerobic $CH_4$



oxidation is substantial, the resulting isotopic fractionation generates measurable vertical and/or
horizontal seawater gradients that can also be used to identify contrasting biogenic $CH_4$ sources
(Leonte et al., 2020). However, the general overlap in isotope compositions of sediment $CH_4$
(e.g. Thornton et al., 2016b; Sapart et al., 2017) can complicate purely isotope-based
determinations of sources.
Measurements of the natural radiocarbon content of dissolved oceanic $CH_4$, while being
highly specialized and requiring substantial amounts of ship time and processing (Kessler and
Reeburgh, 2005; Sparrow and Kessler, 2017), provide valuable source information because the
$^{14}C$-$CH_4$ measurements are normalized to the same $\delta^{13}C$ value and are unaffected by the extent
of oxidation. The bubbles sampled from hydrate and active seafloor seeps are largely devoid of
radiocarbon (Pohlman et al., 2009; Kessler et al., 2008; Douglas et al., 2016). However, $CH_4$ in
sediments can also be derived from more modern or recently deposited organic material and an
exact determination of individual contributions is hard to achieve (Kessler et al., 2008; Sparrow
et al., 2018). The powerful insights made by radiocarbon-$CH_4$ investigations would be further
strengthened by concurrent sampling of other analytes that offer $CH_4$ source information, such as
clumped isotopes. Isotope clumping, the co-occurrence of two or more of the less abundant
isotopes in a molecule (e.g. $^{13}C$ and $^{2}H$ or $^{1}H$ and $^{2}H$), provides unique information on marine
$CH_4$ sources (Stolper et al., 2014; Wang et al, 2015; Douglas et al., 2017; Young et al., 2017;
Giunta et al. 2019). In this approach, the isotopic deviations in samples from their random
probability distributions can give insight into formation temperature and the extent of
biochemical disequilibrium. However, the sample size required for a clumped isotope analysis
in the oceanic environment away from areas of seafloor emission is large and exceeds the
already demanding volume requirements for $^{14}C$ analyses by 1–2 orders of magnitude (Douglas
et al., 2017). While the requirement of large sample size and lengthy measurement time
currently preclude their more widespread application, clumped isotope measurements offer
future promise in refining our understanding of the processes of marine $CH_4$ production and
consumption.

**4. Nitrous oxide in marine environments**

The large-scale spatial distribution of $N_2O$ in the global ocean is reasonably well-established.
The highest open ocean $N_2O$ values are in upwelling environments, where concentrations extend



up to micromolar levels (Arévalo-Martínez et al., 2015) and production rates can be as high as
120 nM d$^{-1}$ (Frey et al., 2020). The highly elevated $N_2O$ concentrations can be proximal to
regions with some of the lowest recorded $N_2O$ concentrations, in the cores of $O_2$ deficient zones.
This coexistence of the highest and lowest observed $N_2O$ concentrations over vertical distances
of tens of meters make upwelling regions a focal point for $N_2O$ research, particularly since $O_2$
deficient ocean zones are increasing in size (Stramma et al., 2011). In contrast, in the surface
waters of the expansive oligotrophic ocean gyres, $N_2O$ is weakly supersaturated (103-105%)
with respect to atmospheric equilibrium (Weiss et al., 1992; Wilson et al., 2017, Charpentier et
al., 2010). Nitrous oxide becomes more highly saturated in the surface waters of equatorial
upwelling regions due to the upward advection of $N_2O$-rich waters (Arévalo-Martínez et al.,
2017). For the Arctic Ocean, the data indicate low net $N_2O$ emissions, with some areas acting as
net $N_2O$ sources and others as $N_2O$ sinks (Fenwick et al., 2017, Zhang et al., 2015).

Several parameters control net $N_2O$ emissions from the ocean, including temperature,

salinity, dissolved $O_2$, apparent oxygen utilization (AOU), nutrients, and microbial community
abundance and composition. A recent modeling study trained with just three of these variables
(chlorophyll, $O_2$, and AOU) accounted for 60% of the observed variability in oceanic $N_2O$
concentrations (Yang et al., 2020), highlighting the importance of $N_2O$ in productive upwelling
systems. Correlations between $N_2O$ and environmental variables provide some insight into the
factors controlling its distribution, but they provide no information about the microorganisms or
metabolic pathways involved. Microbial production of $N_2O$ occurs during the metabolic
processes of nitrification and denitrification (Stein and Yung, 2003). To determine which
process dominates $N_2O$ production at any given location requires the application of multiple
methodological approaches, ideally in parallel.

One of the most commonly used approaches is the incubation of discrete water samples

under *in situ* conditions with stable isotope ($^{15}$N) addition such as $^{15}$N enriched $NH_4^+$, $NO_2^-$ or
$NO_2^-$/ $NO_3^-$ to measure $N_2O$ production rates from nitrification and denitrification, respectively
(e.g. Ji et al., 2017). These approaches also provide insight into the microorganisms involved.
For example, $N_2O$ resulting from archaeal $NH_4^+$ oxidation is mostly formed from a combination
of $NH_4^+$ and another N compound (e.g. $NO_2^-$) whereas bacteria produce $N_2O$ from $NH_4^+$ alone
(Santoro et al., 2011, Stieglmeier et al., 2014; Carini et al. 2018; Lancaster et al., 2018; Frey et
al. 2020). Unfortunately, as with all incubation-based approaches $^{15}$N techniques are subject to



bottle artifacts, and the strong dependence of $N_2O$ production and consumption on ambient $O_2$
increases the potential for contamination during the collection and manipulation of anoxic deep
seawaters.  Incubation based rate measurements are also compromised by abiotic $N_2O$
production via chemodenitrification, specifically the reduction of $NO_2^-$ coupled to $Fe^{2+}$
oxidation, as observed in high Fe environments (Ostrom et al., 2016; Buchwald et al., 2016;
Wankel et al., 2017).  These issues highlight the need for incubation techniques that mitigate the
effect of experimental artifacts (Stewart et al., 2012).

In addition to isotope addition and incubation, natural abundance water-column

measurements of $N_2O$ concentrations, isotopes, and isotopomers yield valuable rate and process
information. These measurements are free from experimental artifacts and can be used to
integrate over appropriate temporal and spatial scales.  For example, nitrification in sunlit waters
has been inferred from $N_2O$ distributions (Dore and Karl, 1996), and $N_2O$ production close to the
ocean surface is a large contributor to the uncertainty in oceanic $N_2O$ emissions (Ward et al.,
1982; Zamora and Oeschlies, 2014).  Isotopomers are isomers having the same number of each
isotope of each element but differing in their structural positions.  Nitrous oxide isotopomers are
increasingly used, sometimes in combination with box models, to estimate the rates of different
$N_2O$ production pathways, in the upwelling systems off southern Africa (Frame et al., 2014) and
Peru (Bourbonnais et al., 2017).  There is however some disagreement about whether isotopomer
signatures are robust indicators of the formation pathway (Yoshida and Toyoda, 2000; Sutka et
al., 2006) or whether there is fractionation during production (Schmidt et al., 2004; Casciotti et
al., 2018).  Greater clarity is therefore required in the use of $N_2O$ isotopes and isotopomers to
infer metabolic pathways of $N_2O$ formation.  Notwithstanding this issue, field measurements of
$N_2O$ isotopes and/or isotopomers have the potential to greatly increase current experimental
capabilities and robustness (Yu et al., 2020).  However, the development of spectroscopic gas
analysis systems that have been so advantageous to $CH_4$ research has been slower for $N_2O$.  This
is due to the higher costs and the increased complexity of the laser systems, although progress is
being made to improve instrumental precision, and to decrease matrix effects and spectral
interferences (e.g. Harris et al., 2019).

A better understanding of the microorganisms responsible for $N_2O$ production and

consumption is fundamental to deriving more accurate estimates of process rates. For example,
the metabolic activity of ammonia oxidizing archaea can exceed that of ammonia oxidizing

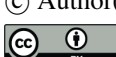



bacteria in the ocean (Santoro et al., 2010; Löscher et al., 2012; Fuchsman et al., 2017). The
differing sensitivities of these archaea and bacteria to dissolved $O_2$ (Stahl and de la Torre, 2012;
Hink et al., 2017) are a critical factor in evaluating the microbial response to changing
environmental conditions, as shown for the terrestrial environment (Prosser at al., 2020).
Therefore, to understand the impact of deoxygenation on oceanic $N_2O$ emission requires a better
understanding of both archaeal and bacterial metabolisms and their environmental niches. Field-
based sequencing not only characterizes the community but can highlight potential metabolic
pathways when they might not otherwise be inferred. For example, transcripts encoding for $N_2O$
consumption (nosZ) have repeatedly been identified in the oxic water column, despite
denitrification being an anaerobic metabolic process (Wyman et al., 2013; Sun et al., 2017). The
transcription of nosZ has been also located in highly dynamic $O_2$ permeable coastal sediments
(Marchant et al., 2017). Denitrification under aerobic conditions is attributed to fluctuations in
$O_2$, $NO_3^-$, organic matter and other parameters that affect the availability of electron donors and
acceptors which ultimately influences whether a coastal environment is a net source or sink of
$N_2O$, as discussed in the next section.

## 358     5. $CH_4$ and $N_2O$ in shallow marine environments

Coastal and other shallow (<50 m) marine systems are globally relevant $CH_4$ and $N_2O$ source
regions. However, their emission rates to the atmosphere are weakly constrained in comparison
with the open ocean. Several factors contribute to the uncertainty, including the high diversity of
coastal and shallow marine ecosystems and lack of consistency in adequately defining them,
locally heterogeneous conditions causing strong spatial and temporal concentration gradients,
highly uncertain spatial distribution of $CH_4$ seeps, a bias towards studies in the northern
hemisphere, and incomplete or sometimes inappropriate sampling strategies (Al-Haj and
Fulweiler, 2020). Until these issues are resolved it will remain difficult to adequately define the
contribution from shallow marine systems to global $CH_4$ and $N_2O$ budgets. An important
illustration of this is reflected in the prevailing view that large geological sources (e.g. seeps,
mud volcanoes, and hydrates) are the main contributors to marine $CH_4$ emissions (Ciais et al.,
2013). The most recent modeled estimate of global marine $CH_4$ emissions (6–12 Tg $CH_4$ $yr^{-1}$)
reported that near-shore environments (depths of 0–50 m) contribute a large and highly uncertain
diffusive flux (Weber et al., 2019). A study of coastal ecosystems, in this case defined as shelf,



estuarine, and tidally influenced rivers, estimated them to contribute 7 Tg $CH_4$ $yr^{-1}$ (Anderson et
al., 2010) while another estimated 1–7 Tg $CH_4$ $yr^{-1}$ for estuaries alone (Borges and Abril, 2011).
Similar uncertainties exist for $N_2O$. Estimates of coastal $N_2O$ emissions (which include coastal,
estuarine, and riverine sources) range from 0.1–2.9 Tg N $yr^{-1}$ (Ciais et al., 2013), although a
recent review of $N_2O$ production across a range of estuarine habitats placed $N_2O$ fluxes at the
lower end of these estimates (0.17–0.95 Tg N $yr^{-1}$) (Murray et al., 2015). Based on these data,
coastal systems account for around one third of total marine $N_2O$ emissions (Yang et al., 2020).
The direct quantification of $CH_4$ and $N_2O$ emissions from shallow coastal ecosystems has
historically involved using gas concentrations measured in discrete water and air samples
combined with a gas transfer velocity ($k_w$). For the coastal and open ocean, the dominant driver
of gas exchange is wind speed (e.g. Nightingale et al., 2000; Wanninkhof, 2014) whereas in
nearshore, shallow water environments the interaction of water, depth, and tidal current speeds
may be a major contributor to near surface turbulence. Several $k_w$ parameterizations are now in
use for coastal waters (e.g. Raymond and Cole 2001; Kremer et al., 2003; Zappa et al., 2003;
Borges and Abril, 2011; Ho et al. 2011; Rosentreter et al., 2017; Jeffrey et al., 2018) which
increases the uncertainties associated with $CH_4$ and $N_2O$ emissions. For example, a fivefold
variation in $CH_4$ emissions from a single system occurred when applying different
parameterizations to the measured gradients in $CH_4$ (Ferrón et al., 2007).
In order to constrain emissions over small areas continuous air-sea fluxes can be measured
using free-floating chambers (e.g. Bahlmann et al., 2015; Rosentreter et al., 2018; Yang et al.,
2018; Murray et al., 2020), but issues related to turbulence modification may still generate flux
artifacts (Upstill-Goddard, 2006). To overcome these problems in the future, a greater reliance
on direct and robust continuous techniques for air-sea flux measurement, such as eddy
covariance (e.g. Podgrajsek et al., 2016) that avoids any need for $k_w$, will be necessary.
Combining this with new analytical techniques such as cavity enhanced absorption spectroscopy
(CEAS) and non-dispersive infrared (NDIR) should continue to improve the quality of such
estimates (McDermitt et al., 2011; Nemitz et al., 2018; Maher et al., 2019). Indeed, eddy flux
towers aboard ships (Thornton et al., 2020) and in coastal locations (Yang et al., 2016; Gutiérrez-
Loza et al., 2019) are now being equipped with $CH_4$ instrumentation that enables the integration
of $CH_4$ fluxes over large areas. There are fewer $N_2O$ flux estimates made with CEAS and NDIR
and the implementation of $N_2O$ sensors on eddy flux towers remains limited. Recently, $N_2O$



emissions from three major Eastern Boundary Upwelling Systems were quantified using atmospheric measurements from coastal monitoring stations highlighting their ability to attain multi-year time-series measurements (Ganesan et al., 2020).

Flux towers at fixed locations provide a stable instrument platform and facilitate the collection of ancillary data such as water-column depth, tidal motions (Rosentreter et al., 2018; Huang et al., 2019; Pfeiffer-Hebert et al., 2019), and other information relating to diel processes (Maher et al., 2016). Such data are important because for example, the magnitude of $CH_4$ and $N_2O$ fluxes vary over a diel period depending on the redox environment as a result of tidal effects and changes in inorganic N and $O_2$ availability (Seitzinger and Kroeze, 1998; Call et al., 2015; Vieillard and Fulweiler, 2014; Maher et al., 2015; Murray et al., 2015; Foster and Fulweiler, 2019). The magnitude of $CH_4$ and $N_2O$ fluxes also varies over longer temporal scales (seasonally to yearly) due to additional factors such as groundwater inputs, adjacent land-use, dissolved $O_2$, organic matter content and quality, and macrofaunal distributions (Barnes and Upstill-Goddard, 2011; Upstill-Goddard and Barnes, 2016; Gelesh et al., 2016; Bonaglia et al., 2017; Borges et al., 2018; Wells et al., 2018; Ray et al., 2019; Al-Haj and Fulweiler, 2020; Reading et al., 2020). To determine the contributing factors and resolve the spatial distributions, mobile sampling platforms such as small vessels (Müller et al., 2016; Brase et al., 2017; Tait et al., 2017), and autonomous vehicles (Manning et al., 2019) are essential. Recent improvements in gas sensors and in technology such as sonar and ebullition sensors will further increase our ability to measure dynamic fluxes (Maher et al., 2019; Lohrberg et al., 2020). Improvements to the quality and quantity of $CH_4$ and $N_2O$ measurements in coastal systems will enable the development of iterative forecast models, further improving estimates of global coastal $CH_4$ and $N_2O$ fluxes.

## 6. Leveraging culture studies to further our ecosystem understanding

A more complete understanding of marine $CH_4$ and $N_2O$ necessitates closer integration between biogeochemistry, model requirements, and targeted microbiological studies involving both single microorganism isolates and enrichment cultures. Marine $CH_4$ and $N_2O$ budgets deriving from both 'bottom-up' (e.g. emissions inventories, ocean and terrestrial process models) and 'top-down' (e.g. inverse analyses of atmospheric trace-gas measurements) approaches would greatly benefit from more highly constrained metabolic processes. Specifically, this includes rates of



$CH_4$ or $N_2O$ production and consumption for key model microorganisms, and the kinetic
parameters associated with these metabolic rates. Reliable inventories of key microbially
mediated process rates will improve the robustness of Earth System models used for predicting
climate-mediated changes to marine $CH_4$ and $N_2O$ emissions.
For $N_2O$, laboratory studies quantifying microbial process rates, such as for nitrification and
denitrification, are relatively few (e.g. Frame and Casciotti 2010; Santoro et al. 2011; Löscher et
al. 2012; Ji et al. 2015; Qin et al., 2017). Consequently, models largely continue to use process
rates optimized using water column concentrations of $N_2O$, $O_2$, and related nitrogen cycle
quantities (e.g. Battaglia and Joos, 2018; Buitenhuis et al., 2018; Landolfi et al., 2017). Future
model parameterizations for $N_2O$ will require information on the variability of microbial process
yields derived from culture studies with controlled varying conditions of $O_2$ (Goreau et al. 1980,
Frame and Casciotti 2010, Löscher et al. 2012; Ji et al., 2018), pH (Breider et al., 2019; Hopkins
et al. 2020), temperature, and nutrients. Automated incubation systems have measured $N_2O$
production kinetics and yield as functions of the concentrations of $O_2$ and total ammonia nitrogen
(Molstad et al., 2007; Hink et al., 2017). Quantifying the physiology of relevant microorganisms
and connecting them to environmental characteristics will provide insights into why, for
example, some shallow marine habitats act as $N_2O$ sinks while others are $N_2O$ sources, or how
$N_2O$ is produced in well oxygenated open-ocean waters, as compared to oxygen deficient zones.
For $CH_4$, a key requirement to relate *in situ* $CH_4$ production with transport to atmospheric
emissions is our ability to accurately determine rates of $CH_4$ oxidation. Fundamental issues
include the challenges of cultivating methanotrophs and of replicating environmental conditions
such as pressure and the chemistry of $CH_4$ gas bubbles. The increased emphasis on $CH_4$
dynamics in shallow water environments highlighted in Section 5, must be supported by culture-
based measurements of $CH_4$ oxidation that control for temperature, $O_2$ and other important
variables. In comparison to $CH_4$ oxidation, culture-based studies are used increasingly to
identify organisms capable of aerobic $CH_4$ production and their underlying metabolic pathways
(Carini et al., 2014; Klintzsch et al; 2019; Bižić et al., 2020).
Specific cellular yields and consumption rates of $CH_4$ and $N_2O$ are not the sole objective of
culturing experiments. Cultivation of microorganisms involved in $CH_4$ and $N_2O$ production and
consumption provides vital information into the physiology, metabolism, and interactions of
environmentally relevant clades. When combined with genomic approaches, insights can



therefore be gained into the diversity and global distribution of organisms involved in $CH_4$ and
$N_2O$ cycling. For $CH_4$ some unexpected physiologies have been revealed (Ettwig et al., 2010;
Haroon et al., 2013; Ettwig et al., 2016), which has directed research into sources and sinks of
$CH_4$ in the natural environment. Similarly, our understanding of how and when ammonia
oxidizers produce $N_2O$ has been facilitated by studies of cultured nitrifiers and detailed analysis
of their biochemistry (Stahl and de la Torre, 2012; Caranto and Lancaster, 2017). Recent
combinations of cultivation studies with environmental genomics, albeit largely for terrestrial
systems, have revealed a variety of denitrifiers, many of which are only involved in specific
denitrification steps (Ganesh et al., 2014; Lycus et al, 2017;  Hallin et al, 2018; Marchant et al.,
2018; Conthe et al, 2019).

**7. Outlook and priorities for marine $CH_4$ and $N_2O$ measurements**
This perspectives article has assessed the collective ability of the scientific community to
determine the spatial variability of marine $CH_4$ and $N_2O$ distributions, the underlying
mechanisms that determine this variability, and the resulting sea-to-air emissions. Shallow
marine environments and oxygen deficient zones are widely recognized as deserving of greater
attention because they have high $CH_4$ and $N_2O$ concentrations with inherently high uncertainties
that complicate any assessment of their emissions to air (Bange et al., 1994; Bange et al., 1996;
Bakker et al., 2014; James et al., 2016; Borges et al., 2016; Tian et al., 2020). Fortunately, recent
technological advances that have increased our ability to conduct high-resolution measurements
allow an optimistic outlook for making substantial progress in quantifying the $CH_4$ and $N_2O$
budgets of these ecosystems. Even so, the inherent complexity of shallow marine environments
clearly warrants a strategically coordinated approach to optimize the value of future studies.
Issues to consider include identifying the locations of complementary sampling sites,
standardizing sampling strategies and techniques, and agreeing the use of common ancillary
measurements that set the broad biogeochemical context (Bange et al., 2019). In contrast to the
open ocean, measurement campaigns in shallow water environments are amenable to the use of
eddy covariance flux towers, and they have the potential to lever resources from existing
observation networks, which in North America include the Long-Term Ecological Research
network (LTER) and the National Estuarine Research Reserve (NERR) System (Novick et al.,
2018). Indeed, such activities are already underway; an increasing number of flux towers are





being equipped for $CH_4$ measurements (Torn et al., 2019) and future efforts should focus on the
inclusion of $N_2O$ (see Section 5).

We are encouraged that the Global Carbon Project with its objective of developing a

complete picture of the global carbon cycle including interactions and feedbacks has expanded to
include $CH_4$ (Saunois et al., 2020) and is now incorporating $N_2O$ (Tian et al., 2020).  These
Projects compile the most recent data from peer-reviewed analyses of the sources and sinks of
atmospheric $CH_4$ and $N_2O$ from both natural and human activities.  For example, the aquatic
components of the recent Global Carbon Project $N_2O$ budget reported emissions from the open
ocean, inland waters, estuaries and coastal zones.  Low-oxygen oceanic regions associated with
eastern-boundary upwelling zones, and the coastal ocean were identified as key regions with
significant $N_2O$ variability requiring more detailed assessment via measurement campaigns and
model analyses (Tian et al., 2020).  Coordinating with global initiatives such as the Global
Carbon Project and identifying other areas of synergistic $CH_4$ and $N_2O$ research of mutual
benefit to oceanographers and scientists studying other biomes serve to strengthen the scientific
achievements of all involved (Ganesan et al., 2019).  Furthermore, as highlighted in Section 6,
field observations alone are insufficient to improve the robustness of Earth System models and
leveraging laboratory-based microbial process studies is highly recommended.

The success of any coordinated $CH_4$ and $N_2O$ research program relies heavily on having

uniformly high confidence in the various resulting datasets and their interoperability, and we
identify three key initiatives that are paramount to ensuring this:

(i) The first is to develop and adopt Standard Operating Protocols (SOPs) to help obtain

intercomparable $CH_4$ and $N_2O$ datasets of the highest possible accuracy and precision.  In our
recent marine $CH_4$ and $N_2O$ inter comparison exercise we concluded that the diversity of
analytical procedures employed by the participants was a major cause of high variability between
the reported concentrations, highlighting an urgent requirement for $CH_4$ and $N_2O$ SOPs (Wilson
et al., 2018). Consequently, these SOPs are now being compiled, and they will be freely
available via the Ocean Best Practices System.

(ii) The second is the regular, routine inter comparison of measurements, by periodically

distributing to the community "consensus material", i.e. water samples in which $CH_4$ and $N_2O$
concentrations are known with high confidence, obtained by pooling analyses from several
laboratories with demonstrated analytical capability.  These will allow us to routinely monitor



data inter comparability and accuracy, particularly in the case of highly elevated concentrations
of $CH_4$ and $N_2O$, i.e. those exceeding atmospheric equilibrium concentrations by at least an order
of magnitude.
(iii) The third activity is increased use and support for MEMENTO. Until now the main
function of MEMENTO has been as a data repository. In this regard, it has been very valuable in
supporting the modeling components of $CH_4$ and $N_2O$ research (see Section 3). We encourage a
much more widespread, routine, use of this data facility, with submitted data produced according
to the SOPs and inter comparison procedures. To maintain its relevance, MEMENTO must
continue to build its activities and develop into an 'ocean $CH_4$ and $N_2O$ Atlas'. The international
marine carbon science community has widely embraced such an approach for $CO_2$, by
submitting data to the Surface Ocean $CO_2$ Atlas (SOCAT), which was initiated in response to the
need for a quality controlled, publicly available, global surface $CO_2$ dataset (e.g. Bakker et al.,
2016). We believe establishing a similar data product for marine $CH_4$ and $N_2O$ to be essential
for supporting future global modeling efforts and to enhance and reward community
engagement.
The benefits of pursuing the three activities described above have already been clearly
demonstrated for carbon system measurements in the ocean. The intercomparability and high
accuracy and precision of carbon system measurements was achieved by streamlining
methodological approaches, universally adopting agreed SOPs, production of reference material,
and following community-driven quality control procedures (Dickson et al., 2007, Dickson et al,
2010). It is encouraging to see the marine $CH_4$ and $N_2O$ community beginning to move in a
similar direction.

*Acknowledgements*: The workshop was held at the University of California Los Angeles Lake
Arrowhead conference center during 28-31 October 2018 (https://web.whoi.edu/methane-
workshop/). We are grateful to all the participants who made valuable scientific contributions to
the workshop and we thank S. Ferrón for critical comments to the manuscript. The workshop
was sponsored by the Ocean Carbon and Biogeochemistry (OCB) Project Office, which is
supported by the U.S. National Science Foundation OCE (1558412) and the National
Aeronautics and Space Administration (NNX17AB17G). The workshop received additional
funding from the Moore Foundation and the Scientific Committee on Ocean Research (SCOR)



which receives funding from the U.S. National Science Foundation (Grant OCE-1840868) and
contributions by additional national SCOR committees.  The Chilean COPAS $N_2O$ time-series
measurements receives financial support from FONDECYT (1200861).

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

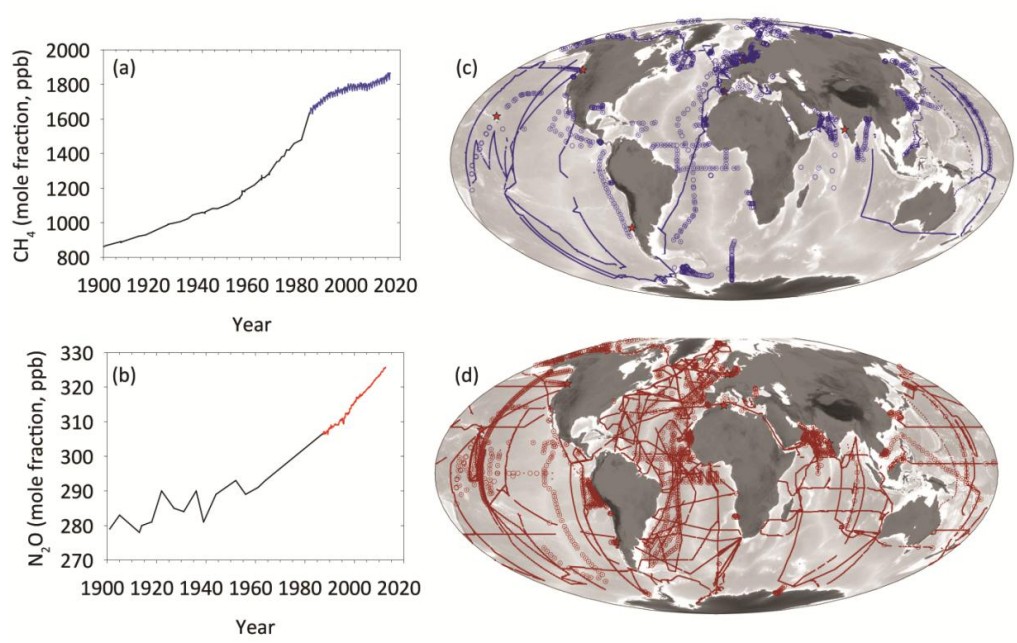


Figure 1. Atmospheric values of (a) CH$_4$ and (b) N$_2$O with the black lines reconstructed from
ice-core measurements (Etheridge et al., 1998; Machida et al., 1995) and the colored lines from
Mauna Loa Observatory (https://www.esrl.noaa.gov/gmd/dv/data/). Global maps of marine (c)
CH$_4$ and (d) N$_2$O measurements available from the MEMENTO database
(https://memento.geomar.de/). The 2018 workshop focused on the marine contribution to
atmospheric CH$_4$ and N$_2$O and the underlying microbial and biogeochemical control
mechanisms.

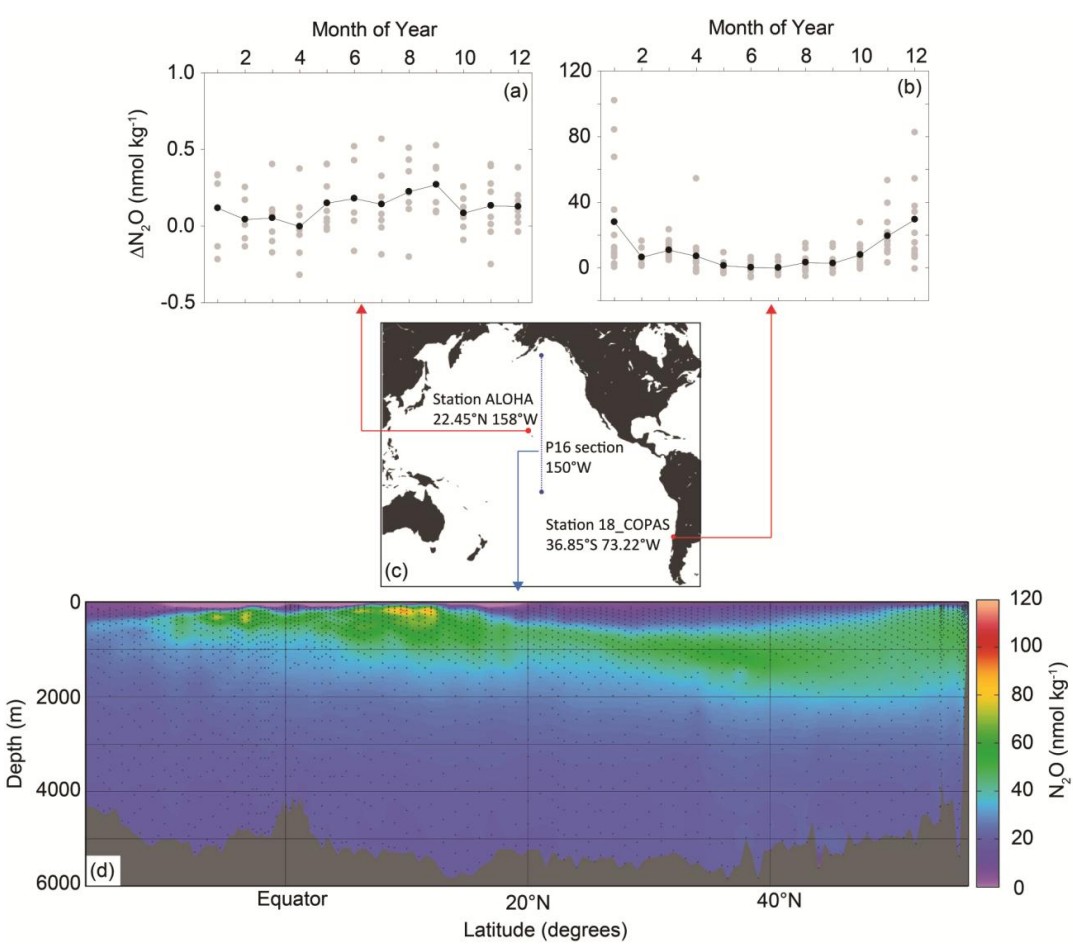


Figure 2. Repeat oceanic observations include both (a, b) fixed location time-series monitoring
observations and (c,d) hydrographic surveys.  Together, such field observation programs helps
resolve temporal variability ranging from months to years and spatial variability at the ocean
basin scale (see Fig. 3).  The Station ALOHA data derive from Wilson et al. (2018), the Station
18 data derive from Farías et al. (2015), and the P16 transect was conducted in 2015 by the
NOAA PMEL Tracer Group as part of the GO-SHIP program.  The data shown in the plots are
$N_2O$ concentrations, either as $\Delta N_2O$ (i.e. deviation from equilibrium value) or absolute values.









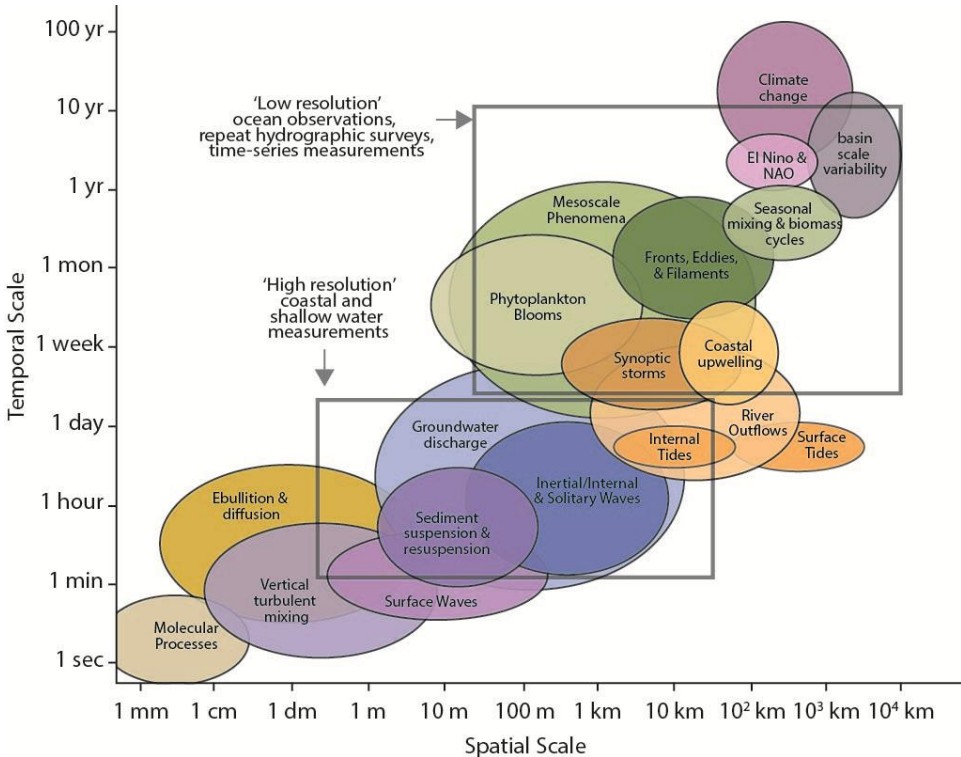


Figure 3. Time-space scale diagram illustrating various physical, biological, and climatological

processes relevant to marine $CH_4$ and $N_2O$ (adapted from Dickey, 2003). To date, the majority

of marine $CH_4$ and $N_2O$ measurements resolve variability at the mesoscale level or higher.

Recent technological developments and the need to resolve concentrations and fluxes in shallow

water environments will increase the number of measurements conducted at the sub mesoscale.

1206

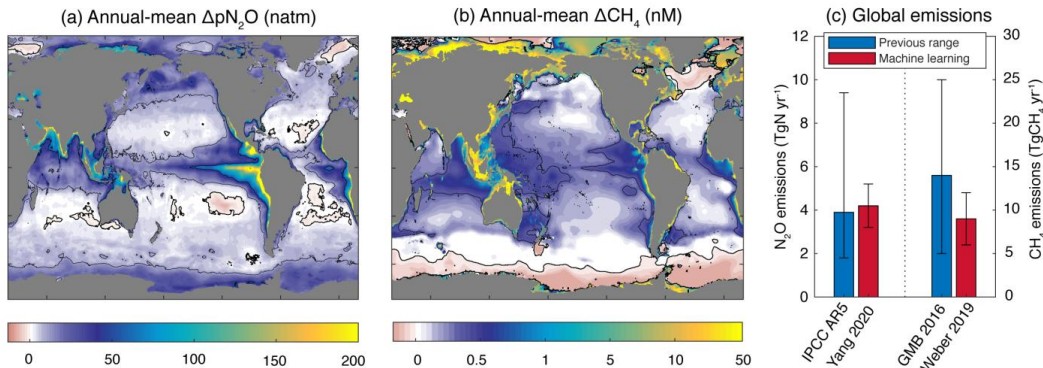

1207

1208

Figure 4. Distributions and emissions of marine CH$_4$ and N$_2$O, (a) Air-sea N$_2$O disequilibrium
mapped using a Regression Forest model (adapted from Yang et al., 2020), (b) Air-sea CH$_4$
disequilibrium mapped using an Artificial Neural Network model (adapted from Weber et al.,
2019). For consistency with the original publications, the air-sea disequilibrium is shown in
different units for N$_2$O (partial pressure) and CH$_4$ (concentration). (c) A summary of global
ocean CH$_4$ and N$_2$O emissions estimated by Yang et al. (2020) and Weber et al. (2019),
compared to the estimates of the IPCC 5th Annual Report (IPCC AR5) and the Global Methane
Budget (Saunois et al., 2016).



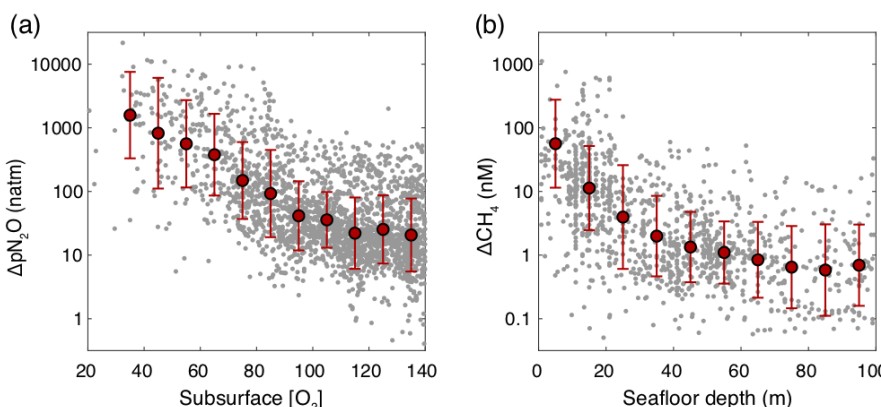


Figure 5. Key environmental predictors of surface ocean $CH_4$ and $N_2O$ gradients. (a) Excess air-
sea $N_2O$ is best predicted by $O_2$ concentrations in the subsurface water-column (base of the
mixed layer to a depth of 100 m) (adapted from Yang et al., 2020). (b) Excess $CH_4$ is best
predicted by seafloor depth, reflecting the supply from anoxic sediments (adapted from Weber et
al., 2019). The grey dots represent individual data points and the red dots with error bars
represent mean ±1s.d. of binned data, using $O_2$ bins of 10 µM width and seafloor depth bins of
10 m width.