# Peer review of "Ideas and perspectives: A strategic assessment of methane and nitrous oxide measurements in the marine environment"

_Biogeosciences, 2020_

## Referee Comment (RC1) · Anonymous Referee #1 · 31 Jul 2020

The ocean is the net source of both CH4 and N2O, which are the second and third largest anthropogenic greenhouse gases. However, the air-sea flux of these gases remains uncertain, due mainly to the lack of sufficient reliable measurement of marine CH4 and N2O. It is thus of urgent need to further strengthen the observation of these greenhouse gases in the ocean. To this end, the authors proposed several perspectives of improving the current observation ability to better constrain and predict the marine CH4 and N2O flux. Overall, I feel these perspectives are essential and clearly stated.

The manuscript has included the main findings of previous researches in this field.

My main concern of the manuscript is that the three initiatives been proposed here are quite similar with the main ideas of the recent study by the authors themselves (i.e., Wilson et al., An intercomparison of oceanic methane and nitrous oxide measurements, 2018; Bange et al., A harmonized nitrous oxide (N2O) ocean observation network for the 21st Century, 2019). I would encourage more specific and further steps of practicing these initiatives, such as providing more detailed plans of developing standard operating protocols, preparing reliable reference gases and samples, planning for regular training exercises... It is also worthwhile to add some ideas for observations, e.g., episodic/ short-term event monitoring (cyclone disturbance, phytoplankton bloom...) and diel rhythm of emission in the coastal zone.

Meanwhile, to better understand and modeling marine CH4 and N2O, process study including molecular and isotope approaches from both lab culture and field study could also be added into the database. In addition to CH4 and N2O observation, the standard measurement of the parameters for air-sea flux calculation, such as the gas transfer velocity or the eddy covariance, should be incorporated to derive accurate air-sea flux.

Finally, given the profound but unclear impacts of the global change and human activities on the marine carbon and nitrogen cycles, research on CH4 and N2O cycling under various external forcing (i.e. deoxygenation, warming, acidification, eutrophication) are encouraged to be incorporated as a component of the database.

The O2 threshold for denitrification is still controversial, the redox potential is likely to be a better index to explore denitrification and other redox reactions relevant to N2O and CH4. In this sense, the measurements of ORP may be included in sampling campaign and database. For modelers, the ORP, which can be connected to electron flow and energy loss-gain, may be useful to advance models with new parameterizations of those chemoautotrophic microorganisms.

The authors synthesized almost all recent documents, which are very useful for beginners who are interested in monitoring marine greenhouse gases. Overall, this is a

well written comprehensive review. Some problems still, many of their statements or illustrations are not referred specifically to the corresponding figures, for example, Fig. 1a, 1b, 2a, 2b, 2c, 2d and 4. Figure 5a is not mentioned in the text.

---

## Referee Comment (RC2) · Anonymous Referee #2 · 17 Aug 2020

The authors here put together a comprehensive review of oceanic CH4 and N2O measurements and state of the knowledge. As an "ideas and prospectives" paper that is lead by the world's best in this field, this contribution is important. This is a timely review with respect to methane especially because of the recent global methane budget which just came out through the global carbon project. In terms of the journal required review criteria: 1. Does the paper address relevant scientific questions within the scope of BG? Yes, the scientific questions they bring up in the review is relevant to BG 2. Does the paper present novel concepts, ideas, tools, or data? As a review paper, they do not bring up novel concepts, but they bring together many concepts in a novel way so I think this paper checks this box. 3. Are substantial conclusions reached? They give

three key initiatives they are pushing with their review. If these initiatives are followed, the scientific consequences for oceanic methane and N2O science to really move forward is clear. 4. Are the scientific methods and assumptions valid and clearly outlined? NA 5. Are the results sufficient to support the interpretations and conclusions? NA 6. Is the description of experiments and calculations sufficiently complete and precise to allow their reproduction by fellow scientists (traceability of results)? NA 7. Do the authors give proper credit to related work and clearly indicate their own new/original contribution? For the most part, yes. 8. Does the title clearly reflect the contents of the paper? yes 9. Does the abstract provide a concise and complete summary? yes 10. Is the overall presentation well structured and clear? yes 11. Is the language fluent and precise? yes 12. Are mathematical formulae, symbols, abbreviations, and units correctly defined and used? NA 13. Should any parts of the paper (text, formulae, figures, tables) be clarified, reduced, combined, or eliminated? no 14. Are the number and quality of references appropriate? For the most part 15. Is the amount and quality of supplementary material appropriate? NA Specific comments: Line 42: reference should be 23, not 22 Line 56: It was good to see the "numerical modeling" portion in the abstract. And modeling came up throughout the review, but it might be more informative to have a section dedicated to what is needed for these models, in a comprehensive way. Specifically, what sort of temporal and spatial resolution is needed? What sort of precision on measurements is required? Line 141: check out: Gelesh, L., et al (2016). Methane concentrations increase in bottom waters during summertime anoxia in the highly eutrophic estuary, Chesapeake Bay, USA. Limnology and Oceanography 61, S253-S266. Line 172: can you be more specific on what predictor variables are for methane and what are for N2O? Just separate the citations here for which gas they focus on and what they find are the predictors Line 190: "other processes". Please elaborate on what processes you mean here. Line 197: check out: Lorensen, T.D., Grienert, J., and Coffin, R.B. (2016). Dissolved methane in the Beaufort Sea and the Arctic Ocean, 1992–2009; sources and atmospheric flux. Limnology and Oceanography 61, 300-323. And, Lapham, L., et al (2017). Dissolved methane concentrations in

the water column and surface sediments of Hanna Shoal and Barrow Canyon, Northern Chukchi Sea. Deep-Sea Research II doi: 10.1016/j.dsr2.2017.01.004. Line 220: check out: Lapham, L., et al (2013). Temporal variability of in situ methane concentrations in gas hydrate-bearing sediments near Bullseye Vent. Geochemistry, Geophysics, Geosystems 14, 2445-2459. Line 242: check out: Grant, N., J., and Whiticar, M.J. (2002). Stable carbon isotopic evidence for methane oxidation in plumes above Hydrate Ridge, Cascadia Oregon Margin. Global biogeochemical cycles 16 (4), 1-13.

---

## Referee Comment (RC3) · Anonymous Referee #3 · 25 Aug 2020

In this paper, Wilson and colleagues provide a blueprint for future research effort into constraining N2O and CH4 emissions from the marine environment. Overall the paper is well written, clear and covers the main points of interest in this research area. One comment that I would make, is that for a perspectives paper on a global issue, the authorship is very USA/Europe heavy. I understand this is a reflection of the OCB workshop attendees, but to ensure global collaboration on this very important issue, engagement with researchers from across the world is needed. While the processes and mechanisms controlling CH4 and N2O production and emission are reasonably well understood, the main issue is a set of SOP and certified reference material to guide the research and provide robust and inter-comparable results.. Engagement

with the broader research community is needed to ensure these best practice protocols are taken up. It is encouraging to read these are currently being developed, but I do wonder how reference material of significant quantities can be produced and delivered to the various labs, particularly those using equilibrator-gas analyzer set ups which are becoming the standard (as opposed to discrete samplers with GC analysis). Some details on how this issue may be overcome would be welcomed.

These are the only minor comments I have on this paper, and I look forward to seeing it in print.

---

## Author Comment (AC1) · 22 Sep 2020

Reviewer #1 The ocean is the net source of both CH4 and N2O, which are the second and third largest anthropogenic greenhouse gases. However, the air-sea flux of these gases remains uncertain, due mainly to the lack of sufficient reliable measurement of marine CH4 and N2O. It is thus of urgent need to further strengthen the observation of these greenhouse gases in the ocean. To this end, the authors proposed several perspectives of improving the current observation ability to better constrain and predict the marine CH4 and N2O flux. Overall, I feel these perspectives are essential and clearly stated. The manuscript has included the main findings of previous researches

in this field.

Authors response »Thank you for your comments

Reviewer comment»My main concern of the manuscript is that the three initiatives been proposed here are quite similar with the main ideas of the recent study by the authors themselves (i.e., Wilson et al., An intercomparison of oceanic methane and nitrous oxide measurements, 2018; Bange et al., A harmonized nitrous oxide (N2O) ocean observation network for the 21st Century, 2019).

Authors response »Reviewer #1 makes the comment that the three initiatives (development of SOPs, intercomparison of seawater samples, and improved usage and output for a centralized data repository) mentioned in Section 7 'Outlook and Priorities' have already been written about in two previous manuscripts. This is a valid comment with regards to the SOPs as their need was first articulated by Wilson et al (2018) and now two years later they are again being advocated for. In our defense, the SOP documents are being produced at this very moment. The work that is currently being undertaken is listed in response to Reviewer #1's next comment. We have now revised the third initiative that previously focused on improved use of a centralized data repository. The third initiative now highlights the need for a Global Data Product for CH4 and N2O. At the moment, the MEMENTO data repository collects CH4 and N2O concentrations which are then used by the modeling community. This activity has been very successful, but it occurs without the publication of any Data Product which would represent a quality controlled synthesis of all the concentration data that have been collected to that point. The absence of a Global Data Product impedes the progress of community-driven CH4 and N2O research on several levels as scientists measuring CH4 and N2O do not receive the appropriate acknowledgement for use of their datasets in Earth system models and there is no common Data Product for the modeling community to use. This situation for CH4 and N2O contrasts sharply with that of pCO2 which releases Global Data Products on an annual basis via Surface Ocean CO2 Atlas initiative. There are much fewer measurements of CH4 and N2O and it is envisaged that a Global Data

Product for CH4 and N2O every 5 years would be sufficient. We have revised the text and Lines 545-557 now read 'The third activity builds on the previous initiative and calls for the production of Global Data Products for dissolved CH4 and N2O measurements. To date, individual CH4 and N2O measurements are represented at the global scale by the MEMENTO database which has been very successful at compiling CH4 and N2O datasets and making them readily accessible to the modeling community. However, the current situation bypasses the important process of compiling a Global Data Product for dissolved CH4 and N2O which represents the public release of accumulated quality controlled datasets. The international marine carbon science community has widely embraced such an approach for fCO2, by submitting data to the Surface Ocean CO2 Atlas (SOCAT), which was initiated in response to the need for a quality controlled, publicly available, global surface CO2 dataset (e.g. Bakker et al., 2016). Due to the fewer measurements, a similar data product for marine CH4 and N2O would be needed every ∼5 years. We consider the production of Global Data Products for dissolved CH4 and N2O to be essential for supporting future global modeling efforts and to enhance and reward community engagement'.

Reviewer comment»I would encourage more specific and further steps of practicing these initiatives, such as providing more detailed plans of developing standard operating protocols, preparing reliable reference gases and samples, planning for regular training exercises.

Authors response »We are happy to inform Reviewer #1 that some these activities have been completed while some are still ongoing. A summary table of all these activities is included below for quick reference. The only activity mentioned by Reviewer #1 that is not currently being planned is cross-training exercises due to the ongoing coronavirus pandemic. Ongoing: Standard Operating Protocols (https://web.whoi.edu/methane-workshop/sops/) and production of consensus material for CH4 and N2O. Completed: Wilson et al. (2018) Intercomparison exercises; Bullister et al. (2016) Production of compressed gas standards; Kock and Bange (2015) Data portal

Reviewer comment»It is also worthwhile to add some ideas for observations, e.g., episodic/ short-term event monitoring (cyclone disturbance, phytoplankton bloom. . .) and diel rhythm of emission in the coastal zone.

Authors response »Reviewer #1 provides some suggestions here for discrete research projects for CH4 and N2O. However, for this overview perspective article, our preference is highlight the availability of analytical tools which can be used to answer any relevant research question and the need for increased coordination among the scientific community. There are multiple examples of this in the text: Application of isotope analysis for methane as mentioned on Lines 241-282 Application of isotopes and isotopomers for nitrous oxide as mentioned on Lines 310-345 Eddy covariance flux towers as mentioned on Lines 499-502 '. . .measurement campaigns in shallow water environments are amenable to the use of eddy covariance flux towers, and they have the potential to lever resources from existing observation networks, which in North America include the Long-Term Ecological Research network (LTER) and the National Estuarine Research Reserve (NERR) System (Novick et al., 2018). Indeed, such activities are already underway; an increasing number of flux towers are being equipped for CH4 measurements (Torn et al., 2019) and future efforts should focus on the inclusion of N2O' Development of mobile sampling platforms as mentioned on Lines 426-428 'To determine the contributing factors and resolve the spatial distributions, mobile sampling platforms such as small vessels (Müller et al., 2016; Brase et al., 2017; Tait et al., 2017), and autonomous vehicles (Manning et al., 2019) are essential'.

Reviewer comment»Meanwhile, to better understand and modeling marine CH4 and N2O, process study including molecular and isotope approaches from both lab culture and field study could also be added into the database.

Authors response »We interpret this comment by Reviewer #1 to suggest that the MEMENTO data archive for CH4 and N2O concentrations could be extended to include other types of datasets including molecular (presumably DNA) and isotopes (presumably natural abundance water-column values). We are strong advocates for data

archival in nationally supported databases. These national data repositories lead the way in making datasets publicly available that adhere to FAIR data principles (Wilkinson et al., 2016). With regards to environmental molecular data, all genetic sequence data should be submitted to national databases (e.g., the National Center for Biotechnology Information (NCBI) GenBank database) that provide access to the most up-to-date comprehensive DNA sequence information. Similarly, water-column isotope and concentration datasets should be submitted to national oceanographic data repositories (e.g. BCO-DMO, BODC). The MEMENTO data archive is a specialized collection of $CH_4$ and $N_2O$ concentrations, and thereby facilitates the use of these trace gas data by the modeling community, as stated on Lines 172-173: "MEMENTO is now sufficiently mature to support descriptions of the broad-scale surface distributions of $CH_4$ and $N_2O$ (e.g. Suntharalingam et al., 2012; Zamora and Oschlies, 2014; Buitenhuis et al., 2018; Battaglia and Joos, 2018)." The datasets should also be deposited in the appropriate national archives to ensure their long-term survival and adherence to FAIR data principles. When submitting data to MEMENTO, there is the option to cross-reference with complementary or co-collected datasets (e.g., DNA or isotope datasets) and also provide a link to publications that include this information. Wilkinson, M.D., Dumontier, M., Aalbersberg, I.J., Appleton, G., Axton, M., Baak, A., Blomberg, N., Boiten, J.W., da Silva Santos, L.B., Bourne, P.E. and Bouwman, J., 2016. The FAIR Guiding Principles for scientific data management and stewardship. Scientific data, 3(1), pp.1-9.

Reviewer comment»In addition to $CH_4$ and $N_2O$ observation, the standard measurement of the parameters for air-sea flux calculation, such as the gas transfer velocity or the eddy covariance, should be incorporated to derive accurate air-sea flux.

Authors response »We interpret this comment by Reviewer #1 to suggest that there should be a uniform application of the gas transfer velocity. However, until there is an understanding of which parameterizations are most suitable for the different coastal environments with their inherently different characteristics of fetch, depth, and tidal currents, this is not possible. This is highlighted on Lines 387-397 of the manuscript.

Reviewer comment»Finally, given the profound but unclear impacts of the global change and human activities on the marine carbon and nitrogen cycles, research on CH4 and N2O cycling under various external forcing (i.e. deoxygenation, warming, acidification, eutrophication) are encouraged to be incorporated as a component of the database.

Authors response »The manuscript mentioned the influences of different stressors in several places. In the Introduction Lines 103-109 state '...the marine environment is susceptible to an accelerating rate of anthropogenic change that will continue to modify the global cycles of carbon and nitrogen into the future. Environmental impacts on marine CH4 and N2O distributions include increasing seawater temperatures, decreasing concentrations of dissolved oxygen (O2), acidification, retreat of ice and mobilization of carbon substrates from former permafrost, altering coastal run-off, and eutrophication (IPCC, 2019)'. In Section 3 on CH4, Lines 220-2223 state 'Seabed CH4 emissions are hypothesized to increase in a warming ocean through the decomposition of gas hydrates, the degradation of subsea permafrost under some high-latitude seas, and the increased biodegradation of sediment carbon (Romanovskii et al., 2005; Biastoch et al., 2011; Ruppel and Kessler, 2017; Borges et al., 2019)'. In Section 4 on N2O, Lines 291-292 state '....make upwelling regions a focal point for N2O research, particularly since O2 deficient ocean zones are increasing in size (Stramma et al., 2011)'.

Reviewer comment»The O2 threshold for denitrification is still controversial, the redox potential is likely to be a better index to explore denitrification and other redox reactions relevant to N2O and CH4. In this sense, the measurements of ORP may be included in sampling campaign and database. For modelers, the ORP, which can be connected to electron flow and energy loss-gain, may be useful to advance models with new parameterizations of those chemoautotrophic microorganisms.

Authors response »Reviewer #1 suggests that measuring the oxidation reduction potential (ORP) of a sample is likely to be more informative than O2 concentrations. ORP measurements are more commonly associated with wastewater and sediments (e.g.

Tumendelger et al 2019; Zhang et al., 2020) rather than the open ocean for several reasons: (1) Its not only O2 concentrations that are useful but related parameters such as Apparent Oxygen Utilization (AOU) which inform about the deviation from theoretical equilibrium; (2) O2 measurements are nearly always included on every hydrographic CTD cast and it is not evident that commercially available ORP sensors can withstand high pressures. Because of these factors, while we agree with Reviewer #1 that the O2 threshold for denitrification is unresolved, we do not feel that ORP measurements represent a significantly better approach. The manuscript advocates for resolving the relationship between N2O and O2 with increased laboratory based studies. Lines 446-452 state 'For N2O, laboratory studies quantifying microbial process rates, such as for nitrification and denitrification, are relatively few (e.g. Frame and Casciotti 2010; Santoro et al. 2011; Löscher et al. 2012; Ji et al. 2015; Qin et al., 2017). Consequently, models largely continue to use process rates optimized using water column concentrations of N2O, O2, and related nitrogen cycle quantities (e.g. Battaglia and Joos, 2018; Buitenhuis et al., 2018; Landolfi et al., 2017). Future model parameterizations for N2O will require information on the variability of microbial process yields derived from culture studies with controlled varying conditions of O2. . .'.. Finally, Reviewer #1 also mentions modeling the flow of electrons but we feel that this is more relevant at the cellular level (e.g. Hink et al 2017) rather than the ecosystem level which is the focus of this manuscript. Tumendelger et al (2019) Methane and nitrous oxide emission from different treatment units of municipal wastewater treatment plants in Southwest Germany. PloS one, 14(1), p.e0209763. Zhang, X., Wang, X., Feng, W., Li, X. and Lu, H., 2020. Investigating COD and Nitrate–Nitrogen Flow and Distribution Variations in the MUCT Process Using ORP as a Control Parameter. ACS omega, 5, 4576-4587. Hink et al (2017) Kinetics of NH3‐oxidation, NO‐turnover, N2O‐production and electron flow during oxygen depletion in model bacterial and archaeal ammonia oxidizers, Environ. Microbiol., 19, 4882–4896.

Reviewer comment»The authors synthesized almost all recent documents, which are very useful for beginners who are interested in monitoring marine greenhouse gases.

Overall, this is a well written comprehensive review.

Authors response »Thank you

Reviewer comment»Some problems still, many of their statements or illustrations are not referred specifically to the corresponding figures, for example, Fig. 1a, 1b, 2a, 2b, 2c, 2d and 4. Figure 5a is not mentioned in the text.

Authors response »All figures are now referenced in the text.

―――――――――――――――――――――

---

## Author Comment (AC2) · 22 Sep 2020

Reviewers comment»Line 42: reference should be 23, not 22

Authors response»Thank you for catching this typographic error – it has now been changed.

Reviewers comment»Line 56: It was good to see the "numerical modeling" portion in the abstract. And modeling came up throughout the review, but it might be more informative to have a section dedicated to what is needed for these models, in a comprehensive way. Specifically, what sort of temporal and spatial resolution is needed?

[Figure]

What sort of precision on measurements is required?

Authors response»Reviewer #2 raises two important questions about the relationship between observations and models. Specifically Reviewer #2 queries (1) the level of temporal-spatial resolution and (2) the analytical uncertainty required to improve and constrain the models. Our response to these questions and instances where the text has been amended is provided below. A useful contextual analysis of analytical uncertainty associated with CH4 and N2O can be provided by consideration of discerning long term trends. The ocean's response to increasing atmospheric concentrations of CH4 and N2O can be discerned over a timescale of 10 and 5 years respectively, with an analytical uncertainty of 1% (assuming all other parameters remain equal). Stated another way, if we wish to determine whether the oceanic inventory of dissolved CH4 and N2O is increasing at the same rate as the atmosphere, we need to wait 10 years for CH4 and 5 years for N2O, with an analytical uncertainty of 1%. This topic was also discussed in Bange et al (2019) which stated 'Detecting inter-annual N2O signals will require a precision of better than 0.02 nmol L-1 (<0.2%)'. We agree with Reviewer #2 that it is important the manuscript reflects the need for high quality CH4 and N2O measurements and the text has been amended to specifically include this. Lines 526-531 now read 'Currently, there is no defined level of analytical uncertainty for CH4 and N2O analysis that would facilitate the establishment of 'high quality' measurements. However, attaining an analytical uncertainty of ≤1% is considered achievable and for context this would permit the ocean's response to the increasing tropospheric CH4 and N2O mole fractions to be resolved on timescales of 10 and 5 years, respectively, assuming all other parameters remain constant'. Achieving a <1% analytical uncertainty would facilitate more accurate inclusion of the mechanisms driving N2O and CH4 cycling in Earth System models, such as the relationship between N2O yields and O2 concentrations. However, care should be taken that the observations are providing the most useful data needed to improve the models. The manuscript already notes this by commenting that increased resolution of N2O emissions in Earth System models would derive from greater constraint of the Michaelis-Menten kinetics associated with N2O production as

a dependent of O2 concentration. The manuscript text states in Section 7 that this could be achieved from laboratory based measurements where Lines 450-454 read 'Future model parameterizations for N2O will require information on the variability of microbial process yields derived from culture studies with controlled varying conditions of O2, pH, temperature, and nutrients'. The situation is different for the coastal environment which is one of the most uncertain and least predictable sources of methane and nitrous oxide. Using methane as an example, methane concentrations can vary by several orders of magnitude across spatial distances ranging from meters to kilometers. For example, Figure 5b shows methane concentrations increasing by at least 100-fold as depth decreases from 100 m to 5 m. In this setting, accumulating sufficient data points along coastal gradients to resolve the spatial distributions becomes a greater priority than achieving the highest possible analytical accuracy. We have amended the manuscript text to better reflect this and the legend for Figure 3 which illustrates the range of spatial-temporal phenomena that influences CH4 and N2O distributions now states on Lines 1226-1228 'The low resolution oceanographic surveys are more likely to achieve a high level of analytical accuracy compared to high resolution coastal measurements, however this is compensated for by high temporal resolution achieved by underway sampling'. Reviewer #2 also queries whether the manuscript should have a section for the modeling work. However, the preference of the authors is to discuss the insights from models and observations together in the context of the different science themes. One of the workshop objectives was to promote closer collaboration between modelers and observationalists in order to create more complementary tools to answer the most pressing scientific questions. Finally, we wish to point out that it is not just the analytical uncertainty in the CH4 and N2O measurements that requires improvement. As noted in the text on Lines 395-397 'a fivefold variation in CH4 emissions from a single system occurred when applying different parameterizations to the measured gradients in CH4 (Ferrón et al., 2007)'.

Reviewers comment»Line 141: check out: Gelesh, L., et al (2016). Methane concentrations increase in bottom waters during summertime anoxia in the highly eutrophic

estuary, Chesapeake Bay, USA. Limnology and Oceanography 61, S253-S266.

Authors response»The manuscript already cites Gelesh et al (2016) in Section 5 'CH4 and N2O in shallow marine environments'. This is our preferred location for the reference rather than long-term time-series observations.

Reviewers comment»Line 172: can you be more specific on what predictor variables are for methane and what are for N2O? Just separate the citations here for which gas they focus on and what they find are the predictors

Authors response»This has now been clarified and Lines 174-178 now read 'Machine-learning mapping also recently identified the various contributions of physical and biogeochemical predictor variables for CH4 (e.g. depth, primary production; Weber et al., 2019) and N2O distributions (chlorophyll, sea surface temperature, apparent oxygen utilization, and mixed-layer depth; Yang et al., 2020).'

Reviewers comment»Line 190: "other processes". Please elaborate on what processes you mean here.

Authors response»We apologize for the ambiguity associated with this sentence. The text has been revised and Lines 192-196 now state 'In the surface waters of tropical and temperate oceans, a number of factors contribute to the low supersaturation of CH4 including direct aerobic production arising from the degradation of methylated sulfur compounds by phytoplankton (Klintzsch et al., 2019) and methyl phosphonate in phosphorus-depleted waters (Karl et al. 2008, Sosa et al., 2020), indirect production via grazing (Schmale et al., 2018) and abiotic photoproduction (Li et al., 2020).

Reviewers comment»Line 197: check out: Lorensen, T.D., Grienert, J., and Coffin, R.B. (2016). Dissolved methane in the Beaufort Sea and the Arctic Ocean, 1992–2009; sources and atmospheric flux. Limnology and Oceanography 61, 300-323. And, Lapham, L., et al (2017). Dissolved methane concentrations in the water column and surface sediments of Hanna Shoal and Barrow Canyon, Northern Chukchi Sea. Deep-

Sea Research II doi: 10.1016/j.dsr2.2017.01.004.

Authors response»The manuscript now includes the Lorenson et al. (2016) and the Lapham et al. (2017) references on Lines 204 and 205, respectively.

Reviewers comment»Line 220: check out: Lapham, L., et al (2013). Temporal variability of in situ methane concentrations in gas hydrate-bearing sediments near Bullseye Vent. Geochemistry, Geophysics, Geosystems 14, 2445-2459.

Authors response»Thank you for the suggestion, this reference has now been included.

Reviewers comment»Line 242: check out: Grant, N., J., and Whiticar, M.J. (2002). Stable carbon isotopic evidence for methane oxidation in plumes above Hydrate Ridge, Cascadia Oregon Margin. Global biogeochemical cycles 16 (4), 1-13.

Authors response»The Grant (2002) reference suggested by Reviewer #2 provides an in depth analysis of stable isotope methane values and concentrations to determine the quantitative fate of methane of entering water-column from the cold seeps of Hydrate Ridge. However, it doesn't include the broader analysis of higher order hydrocarbons which is the point of the text on Lines 246-249 'For example, combining these measurements with the ratio of $CH_4$ to higher order hydrocarbons (e.g. ethene ($C_2H_4$) and ethane ($C_2H_6$)) can be used to infer for example, whether the origin of the $CH_4$ is thermogenic, sub-seafloor, or biogenic within the water column'. The three references we have cited (Whiticar, 1999; Pohlman et al., 2009; Lan et al., 2019) all include the analysis of additional hydrocarbons in order to provide greater contextualization for the origin of methane.

―――――――――――――――――――

---

## Author Response (AR1)

September 2020

Dear Dr. Wilson,

We are pleased to inform you that the open discussion of your following BG manuscript was closed: Title: Ideas and perspectives: A strategic assessment of methane and nitrous oxide measurements in the marine environment; MS No.: bg-2020-270; MS Type: Ideas and perspectives.

No more referee comments and short comments will be accepted. Now the public discussion shall be completed as follows.  You - as the contact author - are requested to individually respond to all referee comments by posting final author comments on behalf of all co-authors no later than 25 Sept 2020 (final response phase). Please note that your revised manuscript should not be prepared at this stage.

The editorial support team

Copernicus Publications
* * *
September 2020

To the editorial support team at Copernicus Publications.

Thank you for the opportunity to respond to the Reviewer's comments.  We would particularly like to thank the three Reviewers for their thoughtful and constructive comments.  We have included a point-by-point response on the following pages and revised the manuscript as appropriate.

Yours sincerely,

Sam Wilson, on behalf of all authors
* * *
**Reviewer #1**
**The ocean is the net source of both CH4 and N2O, which are the second and third largest anthropogenic greenhouse gases. However, the air-sea flux of these gases remains uncertain, due mainly to the lack of sufficient reliable measurement of marine CH4 and N2O. It is thus of urgent need to further strengthen the observation of these greenhouse gases in the ocean. To this end, the authors proposed several perspectives of improving the current observation ability to better constrain and predict the marine CH4 and N2O flux. Overall, I feel these perspectives are essential and clearly stated.  The manuscript has included the main findings of previous researches in this field.**
Thank you for your comments

**My main concern of the manuscript is that the three initiatives been proposed here are quite similar with the main ideas of the recent study by the authors themselves (i.e., Wilson et al., An intercomparison of oceanic methane and nitrous oxide measurements, 2018; Bange et al., A harmonized nitrous oxide (N2O) ocean observation network for the 21st Century, 2019).**
Reviewer #1 makes the comment that the three initiatives (development of SOPs, intercomparison of seawater samples, and improved usage and output for a centralized data repository) mentioned in Section 7 'Outlook and Priorities' have already been written about in two previous manuscripts.  This is a valid comment with regards to the SOPs as their need was first articulated by Wilson et al (2018) and now two years later they are again being advocated for.  In our defense, the SOP documents are being produced at this very moment.  The work that is currently being undertaken is listed in response to Reviewer #1's next comment.

We have now revised the third initiative that previously focused on improved use of a centralized data repository.  The third initiative now highlights the need for a Global Data Product for $CH_4$ and $N_2O$. At the moment, the MEMENTO data repository collects $CH_4$ and $N_2O$ concentrations which are then used by the modeling community.  This activity has been very successful, but it occurs without the publication of any Data Product which would represent a quality controlled synthesis of all the concentration data that have been collected to that point. The absence of a Global Data Product impedes the progress of community-driven $CH_4$ and $N_2O$ research on several levels as scientists measuring $CH_4$ and $N_2O$ do not receive the appropriate acknowledgement for use of their datasets in Earth system models and there is no common Data Product for the modeling community to use.  This situation for $CH_4$ and $N_2O$ contrasts sharply with that of $pCO_2$ which releases Global Data Products on an annual basis via Surface Ocean $CO_2$ Atlas initiative.  There are much fewer measurements of $CH_4$ and $N_2O$ and it is envisaged that a Global Data Product for $CH_4$ and $N_2O$ every 5 years would be sufficient.

We have revised the text and Lines 545-557 now read '*The third activity builds on the previous initiative and calls for the production of Global Data Products for dissolved $CH_4$ and $N_2O$ measurements. To date, individual $CH_4$ and $N_2O$ measurements are represented at the global scale by the MEMENTO database which has been very successful at compiling $CH_4$ and $N_2O$ datasets and making them readily accessible to the modeling community.  However, the current situation bypasses the important process of compiling a Global Data Product for dissolved $CH_4$ and $N_2O$ which represents the public release of accumulated quality controlled datasets.  The international marine carbon science community has widely embraced such an approach for $fCO_2$, by submitting data to the Surface Ocean $CO_2$ Atlas (SOCAT), which was initiated in response to the need for a quality controlled, publicly available, global surface $CO_2$ dataset (e.g. Bakker et al., 2016).  Due to the fewer measurements, a similar data product for marine $CH_4$ and $N_2O$ would be needed every ~5 years.  We consider the production of Global Data Products for dissolved $CH_4$ and $N_2O$ to be essential for supporting future global modeling efforts and to enhance and reward community engagement*'.

**I would encourage more specific and further steps of practicing these initiatives, such as providing more detailed plans of developing standard operating protocols, preparing reliable reference gases and samples, planning for regular training exercises.**

We are happy to inform Reviewer #1 that some these activities have been completed while some are still ongoing.  A summary table of all these activities is included below for quick reference.  The only activity mentioned by Reviewer #1 that is not currently being planned is cross-training exercises due to the ongoing coronavirus pandemic.

| Activity | Date | Reference | Comments |
|---|---|---|---|
| Intercomparison excercises | 2014-2015 | Wilson et al. (2018) | Seawater collected from the Pacific Ocean and the Baltic Sea was distributed to twenty laboratories |
| Production of compressed gas standards | 2015-2017 | Bullister et al. (2016) | Two gaseous standards (low and high concentration) were shipped to twelve laboratories worldwide. |
| Establishing a common data portal | 2009 onwards | Kock and Bange (2015) | The MEMENTO database provides an archive of $CH_4$ and $N_2O$ datasets. |
| Community building workshop | 2018 | This manuscript | Sixty international scientists participated in the 2018 workshop which outlined the foundations for future activity |

| Standard Operating Protocols | In preparation | https://web.whoi.edu/methane-workshop/sops/ | Nine SOPs are being drafted for publication of the Ocean Best Practice network |
|---|---|---|---|
| Production of consensus material for CH$_4$ and N$_2$O. | n/a | Currently being planned | This is being planned at the moment and a proposal has been submitted to NSF to fund this activity |

**It is also worthwhile to add some ideas for observations, e.g., episodic/ short-term event monitoring (cyclone disturbance, phytoplankton bloom. . .) and diel rhythm of emission in the coastal zone.**
Reviewer #1 provides some suggestions here for discrete research projects for CH4 and N2O.  However, for this overview perspective article, our preference is highlight the availability of analytical tools which can be used to answer any relevant research question and the need for increased coordination among the scientific community.  There are multiple examples of this in the text:

Application of isotope analysis for methane as mentioned on Lines 241-282

Application of isotopes and isotopomers for nitrous oxide as mentioned on Lines 310-345

Eddy covariance flux towers as mentioned on Lines 499-502 '*…measurement campaigns in shallow water environments are amenable to the use of eddy covariance flux towers, and they have the potential to lever resources from existing observation networks, which in North America include the Long-Term Ecological Research network (LTER) and the National Estuarine Research Reserve (NERR) System (Novick et al., 2018).  Indeed, such activities are already underway; an increasing number of flux towers are being equipped for CH4 measurements (Torn et al., 2019) and future efforts should focus on the inclusion of N2O'*

Development of mobile sampling platforms as mentioned on Lines 426-428 '*To determine the contributing factors and resolve the spatial distributions, mobile sampling platforms such as small vessels (Müller et al., 2016; Brase et al., 2017; Tait et al., 2017), and autonomous vehicles (Manning et al., 2019) are essential*'.

**Meanwhile, to better understand and modeling marine CH4 and N2O, process study including molecular and isotope approaches from both lab culture and field study could also be added into the database.**
We interpret this comment by Reviewer #1 to suggest that the MEMENTO data archive for CH$_4$ and N$_2$O concentrations could be extended to include other types of datasets including molecular (presumably DNA) and isotopes (presumably natural abundance water-column values).

 We are strong advocates for data archival in nationally supported databases. These national data repositories lead the way in making datasets publicly available that adhere to FAIR data principles (Wilkinson et al., 2016). With regards to environmental molecular data, all genetic sequence data should be submitted to national databases (e.g., the National Center for Biotechnology Information (NCBI) GenBank database) that provide access to the most up-to-date comprehensive DNA sequence information. Similarly, water-column isotope and concentration datasets should be submitted to national oceanographic data repositories (e.g. BCO-DMO, BODC).

 The MEMENTO data archive is a specialized collection of CH$_4$ and N$_2$O concentrations, and thereby facilitates the use of these trace gas data by the modeling community, as stated on Lines 172-173: "MEMENTO is now sufficiently mature to support descriptions of the broad-scale surface distributions of CH$_4$ and N$_2$O (e.g. Suntharalingam et al., 2012; Zamora and Oschlies, 2014; Buitenhuis et al., 2018; Battaglia and Joos, 2018)." The datasets should also be deposited in the appropriate national archives to ensure their long-term survival and adherence to FAIR data principles. When submitting data to MEMENTO, there is the option to cross-reference with complementary or co-collected datasets (e.g., DNA or isotope datasets) and also provide a link to publications that include this information.

Wilkinson, M.D., Dumontier, M., Aalbersberg, I.J., Appleton, G., Axton, M., Baak, A., Blomberg, N., Boiten, J.W., da Silva Santos, L.B., Bourne, P.E. and Bouwman, J., 2016. The FAIR Guiding Principles for scientific data management and stewardship. Scientific data, 3(1), pp.1-9.

**In addition to CH4 and N2O observation, the standard measurement of the parameters for air-sea flux calculation, such as the gas transfer velocity or the eddy covariance, should be incorporated to derive accurate air-sea flux.**
We interpret this comment by Reviewer #1 to suggest that there should be a uniform application of the gas transfer velocity.  However, until there is an understanding of which parameterizations are most suitable for the different coastal environments with their inherently different characteristics of fetch, depth, and tidal currents, this is not possible.

**Finally, given the profound but unclear impacts of the global change and human activities on the marine carbon and nitrogen cycles, research on CH4 and N2O cycling under various external forcing (i.e. deoxygenation, warming, acidification, eutrophication) are encouraged to be incorporated as a component of the database.**
The manuscript mentioned the influences of different stressors in several places.

In the Introduction Lines 103-109 state '…*the marine environment is susceptible to an accelerating rate of anthropogenic change that will continue to modify the global cycles of carbon and nitrogen into the future.  Environmental impacts on marine $CH_4$ and $N_2O$ distributions include increasing seawater temperatures, decreasing concentrations of dissolved oxygen ($O_2$), acidification, retreat of ice and mobilization of carbon substrates from former permafrost, altering coastal run-off, and eutrophication (IPCC, 2019)*'.

In Section 3 on $CH_4$, Lines 220-2223 state '*Seabed $CH_4$ emissions are hypothesized to increase in a warming ocean through the decomposition of gas hydrates, the degradation of subsea permafrost under some high-latitude seas, and the increased biodegradation of sediment carbon (Romanovskii et al., 2005; Biastoch et al., 2011; Ruppel and Kessler, 2017; Borges et al., 2019)*'.

In Section 4 on $N_2O$, Lines 291-292 state '….*make upwelling regions a focal point for $N_2O$ research, particularly since $O_2$ deficient ocean zones are increasing in size (Stramma et al., 2011)*'.

**The O2 threshold for denitrification is still controversial, the redox potential is likely to be a better index to explore denitrification and other redox reactions relevant to N2O and CH4. In this sense, the measurements of ORP may be included in sampling campaign and database. For modelers, the ORP, which can be connected to electron flow and energy loss-gain, may be useful to advance models with new parameterizations of those chemoautotrophic microorganisms.**
Reviewer #1 suggests that measuring the oxidation reduction potential (ORP) of a sample is likely to be more informative than $O_2$ concentrations.  ORP measurements are more commonly associated with wastewater and sediments (e.g. Tumendelger et al 2019; Zhang et al., 2020) rather than the open ocean for several reasons: (1) Its not only $O_2$ concentrations that are useful but related parameters such as Apparent Oxygen Utilization (AOU) which inform about the deviation from theoretical equilibrium; (2) $O_2$ measurements are nearly always included on every hydrographic CTD cast and it is not evident that commercially available ORP sensors can withstand high pressures.  Because of these factors, while we agree with Reviewer #1 that the $O_2$ threshold for denitrification is unresolved, we do not feel that ORP measurements represent a significantly better approach. The manuscript advocates for resolving the relationship between $N_2O$ and $O_2$ with increased laboratory based studies. Lines 446-452 state '*For $N_2O$, laboratory studies quantifying microbial process rates, such as for nitrification and denitrification, are relatively few (e.g. Frame and Casciotti 2010; Santoro et al. 2011; Löscher et al. 2012; Ji et al. 2015; Qin et al., 2017). Consequently, models largely continue to use process rates optimized using water column concentrations of $N_2O$, $O_2$, and related nitrogen cycle quantities (e.g. Battaglia and Joos, 2018; Buitenhuis et al., 2018; Landolfi et al., 2017). Future model parameterizations for $N_2O$ will require information on the variability of microbial process yields derived from culture studies with controlled varying conditions of $O_2$...'*..

Finally, Reviewer #1 also mentions modeling the flow of electrons but we feel that this is more relevant at the cellular level (e.g. Hink et al 2017) rather than the ecosystem level which is the focus of this manuscript.

Tumendelger et al (2019) Methane and nitrous oxide emission from different treatment units of municipal wastewater treatment plants in Southwest Germany. PloS one, 14(1), p.e0209763.

Zhang, X., Wang, X., Feng, W., Li, X. and Lu, H., 2020. Investigating COD and Nitrate–Nitrogen Flow and Distribution Variations in the MUCT Process Using ORP as a Control Parameter. ACS omega, 5, 4576-4587.

Hink et al (2017) Kinetics of NH3-oxidation, NO-turnover, N2O-production and electron flow during oxygen depletion in model bacterial and archaeal ammonia oxidizers, Environ. Microbiol., 19, 4882–4896.

**The authors synthesized almost all recent documents, which are very useful for beginners who are interested in monitoring marine greenhouse gases. Overall, this is a well written comprehensive review.**
Thank you

**Some problems still, many of their statements or illustrations are not referred specifically to the corresponding figures, for example, Fig. 1a, 1b, 2a, 2b, 2c, 2d and 4. Figure 5a is not mentioned in the text.**
All figures are now referenced in the text.
* * *
**Reviewer #2**
**The authors here put together a comprehensive review of oceanic CH4 and N2O measurements and state of the knowledge. As an "ideas and prospectives" paper that is lead by the world's best in this field, this contribution is important. This is a timely review with respect to methane especially because of the recent global methane budget which just came out through the global carbon project. In terms of the journal required review criteria:**
**1. Does the paper address relevant scientific questions within the scope of BG? Yes, the scientific questions they bring up in the review is relevant to BG**
**2. Does the paper present novel concepts, ideas, tools, or data? As a review paper, they do not bring up novel concepts, but they bring together many concepts in a novel way so I think this paper checks this box.**
**3. Are substantial conclusions reached? They give three key initiatives they are pushing with their review. If these initiatives are followed, the scientific consequences for oceanic methane and N2O science to really move forward is clear.**
**4. Are the scientific methods and assumptions valid and clearly outlined? NA**
**5. Are the results sufficient to support the interpretations and conclusions? NA**

**6. Is the description of experiments and calculations sufficiently complete and precise to allow their reproduction by fellow scientists (traceability of results)? NA**

**7. Do the authors give proper credit to related work and clearly indicate their own new/original contribution? For the most part, yes.**

**8. Does the title clearly reflect the contents of the paper? yes**

**9. Does the abstract provide a concise and complete summary? yes**

**10. Is the overall presentation well structured and clear? yes**

**11. Is the language fluent and precise? yes**

**12. Are mathematical formulae, symbols, abbreviations, and units correctly defined and used? NA**

**13. Should any parts of the paper (text, formulae, figures, tables) be clarified, reduced, combined, or eliminated? no**

**14. Are the number and quality of references appropriate? For the most part**

**15. Is the amount and quality of supplementary material appropriate? NA**

**Specific comments:**

**Line 42: reference should be 23, not 22**

Thank you for catching this typographic error – it has now been changed.

**Line 56: It was good to see the "numerical modeling" portion in the abstract. And modeling came up throughout the review, but it might be more informative to have a section dedicated to what is needed for these models, in a comprehensive way. Specifically, what sort of temporal and spatial resolution is needed? What sort of precision on measurements is required?**

Reviewer #2 raises two important questions about the relationship between observations and models. Specifically Reviewer #2 queries (1) the level of temporal-spatial resolution and (2) the analytical uncertainty required to improve and constrain the models.  Our response to these questions and instances where the text has been amended is provided below.

A useful contextual analysis of analytical uncertainty associated with $CH_4$ and $N_2O$ can be provided by consideration of discerning long term trends.  The ocean's response to increasing atmospheric concentrations of $CH_4$ and $N_2O$ can be discerned over a timescale of 10 and 5 years respectively, with an analytical uncertainty of 1% (assuming all other parameters remain equal).  Stated another way, if we wish to determine whether the oceanic inventory of dissolved $CH_4$ and $N_2O$ is increasing at the same rate as the atmosphere, we need to wait 10 years for $CH_4$ and 5 years for $N_2O$, with an analytical uncertainty of 1%.  This topic was also discussed in Bange et al (2019) which stated '*Detecting inter-annual $N_2O$ signals will require a precision of better than 0.02 nmol $L^{-1}$ (<0.2%)*'. We agree with Reviewer #2 that it is important the manuscript reflects the need for high quality $CH_4$ and $N_2O$ measurements and the text has been amended to specifically include this. Lines 526-531 now read '*Currently, there is no defined level of analytical uncertainty for $CH_4$ and $N_2O$ analysis that would facilitate the establishment of 'high quality' measurements.  However, attaining an analytical uncertainty of ≤1% is considered achievable and for context this would permit the ocean's response to the increasing tropospheric $CH_4$ and $N_2O$ mole fractions to be resolved on timescales of 10 and 5 years, respectively, assuming all other parameters remain constant*'.  Achieving a <1% analytical uncertainty would facilitate more accurate inclusion of the mechanisms driving $N_2O$ and $CH_4$ cycling in Earth System models, such as the relationship between $N_2O$ yields and $O_2$ concentrations.  However, care should be taken that the observations are providing the most useful data needed to improve the models.   The manuscript already notes this by commenting that increased resolution of $N_2O$ emissions in Earth System models would derive from greater constraint of the Michaelis-Menten kinetics associated with $N_2O$ production as a dependent of $O_2$ concentration.  The manuscript text states in Section 7 that this could be achieved from laboratory based measurements where Lines 450-454 read '*Future model parameterizations for*

*$N_2O$ will require information on the variability of microbial process yields derived from culture studies with controlled varying conditions of $O_2$, pH, temperature, and nutrients'.*

The situation is different for the coastal environment which is one of the most uncertain and least predictable sources of methane and nitrous oxide. Using methane as an example, methane concentrations can vary by several orders of magnitude across spatial distances ranging from meters to kilometers. For example, Figure 5b shows methane concentrations increasing by at least 100-fold as depth decreases from 100 m to 5 m. In this setting, accumulating sufficient data points along coastal gradients to resolve the spatial distributions becomes a greater priority than achieving the highest possible analytical accuracy. We have amended the manuscript text to better reflect this and the legend for Figure 3 which illustrates the range of spatial-temporal phenomena that influences $CH_4$ and $N_2O$ distributions now states on Lines 1226-1228 '*The low resolution oceanographic surveys are more likely to achieve a high level of analytical accuracy compared to high resolution coastal measurements, however this is compensated for by high temporal resolution achieved by underway sampling'.*

Reviewer #2 also queries whether the manuscript should have a section for the modeling work. However, the preference of the authors is to discuss the insights from models and observations together in the context of the different science themes. One of the workshop objectives was to promote closer collaboration between modelers and observationalists in order to create more complementary tools to answer the most pressing scientific questions.

Finally, we wish to point out that it is not just the analytical uncertainty in the $CH_4$ and $N_2O$ measurements that requires improvement. As noted in the text on Lines 395-397 '*a fivefold variation in $CH_4$ emissions from a single system occurred when applying different parameterizations to the measured gradients in $CH_4$ (Ferrón et al., 2007)'.*

**Line 141: check out: Gelesh, L., et al (2016). Methane concentrations increase in bottom waters during summertime anoxia in the highly eutrophic estuary, Chesapeake Bay, USA. Limnology and Oceanography 61, S253-S266.**
The manuscript already cites Gelesh et al (2016) in Section 5 'CH4 and N2O in shallow marine environments'. This is our preferred location for the reference rather than long-term time-series observations.

**Line 172: can you be more specific on what predictor variables are for methane and what are for N2O? Just separate the citations here for which gas they focus on and what they find are the predictors**
This has now been clarified and Lines 174-178 now read '*Machine-learning mapping also recently identified the various contributions of physical and biogeochemical predictor variables for $CH_4$ (e.g. depth, primary production; Weber et al., 2019) and $N_2O$ distributions (chlorophyll, sea surface temperature, apparent oxygen utilization, and mixed-layer depth; Yang et al., 2020).'*

**Line 190: "other processes". Please elaborate on what processes you mean here.**
We apologize for the ambiguity associated with this sentence. The text has been revised and Lines 192-196 now state '*In the surface waters of tropical and temperate oceans, a number of factors contribute to the low supersaturation of $CH_4$ including direct aerobic production arising from the degradation of methylated sulfur compounds by phytoplankton (Klintzsch et al., 2019) and methyl phosphonate in phosphorus-depleted waters (Karl et al. 2008, Sosa et al., 2020), indirect production via grazing (Schmale et al., 2018) and abiotic photoproduction (Li et al., 2020).*

**Line 197: check out: Lorensen, T.D., Grienert, J., and Coffin, R.B. (2016). Dissolved methane in the Beaufort Sea and the Arctic Ocean, 1992–2009; sources and atmospheric flux. Limnology and**

**Oceanography 61, 300-323. And, Lapham, L., et al (2017). Dissolved methane concentrations in the water column and surface sediments of Hanna Shoal and Barrow Canyon, Northern Chukchi Sea. Deep-Sea Research II doi: 10.1016/j.dsr2.2017.01.004.**

The manuscript now includes the Lorenson et al. (2016) and the Lapham et al. (2017) references on Lines 204 and 205, respectively.

**Line 220: check out: Lapham, L., et al (2013). Temporal variability of in situ methane concentrations in gas hydrate-bearing sediments near Bullseye Vent. Geochemistry, Geophysics, Geosystems 14, 2445-2459.**

Thank you for the suggestion, this reference has now been included.

**Line 242: check out: Grant, N., J., and Whiticar, M.J. (2002). Stable carbon isotopic evidence for methane oxidation in plumes above Hydrate Ridge, Cascadia Oregon Margin. Global biogeochemical cycles 16 (4), 1-13.**

The Grant (2002) reference suggested by Reviewer #2 provides an in depth analysis of stable isotope methane values and concentrations to determine the quantitative fate of methane of entering water-column from the cold seeps of Hydrate Ridge.  However, it doesn't include the broader analysis of higher order hydrocarbons which is the point of the text on Lines 246-249 '*For example, combining these measurements with the ratio of CH4 to higher order hydrocarbons (e.g. ethene (C2H4) and ethane (C2H6)) can be used to infer for example, whether the origin of the CH4 is thermogenic, sub-seafloor, or biogenic within the water column*'.  The three references we have cited (Whiticar, 1999; Pohlman et al., 2009; Lan et al., 2019) all include the analysis of additional hydrocarbons in order to provide greater contextualization for the origin of methane.
* * *
**Reviewer #3**
**In this paper, Wilson and colleagues provide a blueprint for future research effort into constraining N2O and CH4 emissions from the marine environment. Overall the paper is well written, clear and covers the main points of interest in this research area.**

Thank you for these comments

**One comment that I would make, is that for a perspectives paper on a global issue, the authorship is very USA/Europe heavy. I understand this is a reflection of the OCB workshop attendees, but to ensure global collaboration on this very important issue, engagement with researchers from across the world is needed.**

We thank Reviewer #3 for highlighting the international representation of the authors host countries. The composition of the authors largely derives from participation in the October 2018 workshop, which was sponsored by the US OCB program.  To facilitate participation in the workshop by non-US scientists, we secured funding from the Moore Foundation and to a lesser extent, SCOR.  The number of participants from non-US and non-European countries was Chile (2), Canada (3), South Africa (1), and China (2).  All workshop participants were invited to contribute to the manuscript.  Overall, the workshop and accompanying manuscript tried to attain a balance of male/female, early/senior, and international representation.  A primary goal of this workshop and its products was to identify research priorities and strengthen collaborations across the community.  We agree that the workshop and accompanying manuscript represent only a fraction of the international research community conducting $CH_4$ and $N_2O$ measurements and we will seek to further engage researchers across all nations as we move forward.  For example, the Standard Operating Protocols (SOPs) are currently being written and draft documents will be posted to the website https://web.whoi.edu/methane-workshop/ for community input prior to publication.  Their existence will be announced via the OCB and international partner program newsletters, websites, and social media feeds.  Also, a proposal was submitted to produce consensus material for dissolved methane and nitrous oxide in 2021, which will form the basis for another intercomparison exercise.  We welcome the participation of scientists from all countries in both of these capacity building endeavors.

**While the processes and mechanisms controlling CH4 and N2O production and emission are reasonably well understood, the main issue is a set of SOP and certified reference material to guide the research and provide robust and inter-comparable results.  Engagement with the broader research community is needed to ensure these best practice protocols are taken up. It is encouraging to read these are currently being developed, but I do wonder how reference material of significant quantities can be produced and delivered to the various labs, particularly those using equilibrator-gas analyzer set ups which are becoming the standard (as opposed to discrete samplers with GC analysis). Some details on how this issue may be overcome would be welcomed.**

Reviewer #3 brings up several topics in this comment. As mentioned in response to Reviewer #1, the SOPs are being written and they will be posted to the website https://web.whoi.edu/methane-workshop/ prior to uploading to the Oceans Best Practice Network.  We would like to point out to Reviewer #3 that the 'consensus material' that will be produced for $CH_4$ and $N_2O$ does not meet the necessary criteria to be classified as 'reference material'.  The working definition of Consensus Material is '*Material with properties of a communally agreed value better than 1%, as measured by multiple laboratories*', while reference material is '*Material whose properties are sufficiently established so that it can be used for the calibration of an instrument or the assignment of values to samples*'. Finally, the consensus material is primarily intended to help with the analysis of discrete samples, not equilibrator systems.  This does not mean that calibration of equilibrator systems for $CH_4$ and $N_2O$ cannot be achieved with the help of consensus material.  Indeed one of the SOPs (SOP#7: Underway system) specifically mentions the evaluation of equilibrator systems using discrete samples.

**These are the only minor comments I have on this paper, and I look forward to seeing it in print.**

Many thanks

[revised manuscript text omitted]
 also recently identified the various contributions of physical and biogeochemical predictor variables for $CH_4$

(e.g. depth, primary production; Weber et al., 2019; Fig 4b) and $N_2O$ distributions (e.g.

chlorophyll, sea surface temperature, apparent oxygen utilization, and mixed-layer depth; Yang et al., 2020; Fig. 4a).  The application of gas transfer algorithms to the extrapolated oceanic $CH_4$

and $N_2O$ distributions helped decrease the uncertainty in estimates of global air-sea exchange fluxes (Fig. 4c), thereby fulfilling one of the key goals of MEMENTO (Bange et al., 2009).  Net global open ocean emissions of $N_2O$ are now similarly estimated at 3–5 Tg N $yr^{-1}$ by both Yang et al. (2020) and the Global Nitrous Oxide Project (Tian et al., 2020).  In comparison, net global ocean $CH_4$ emissions from machine-learning mapping were estimated at 6–12 Tg $CH_4$ $yr^{-1}$

(Weber et al., 2019), compared to 9–22 Tg $CH_4$ $yr^{-1}$ in the most up-to-date $CH_4$ synthesis (Saunois et al., 2020).  However, the narrower range for machine-learning derived $CH_4$

emissions retains high uncertainty in regions such as the Arctic, where emissions are highly heterogeneous and compounded by seasonal ice cover.  Identifying the causes for uncertainty in high emission regions will greatly aid future sampling campaigns, as is discussed in the following sections.

**3. Methane in marine environments**

In the surface waters of tropical and temperate oceans, a number of factors contribute to the low supersaturation of $CH_4$ including direct aerobic production arising from the degradation of methylated sulfur compounds by phytoplankton (Klintzsch et al., 2019) and methyl phosphonate in phosphorus-depleted waters (Karl et al. 2008, Sosa et al., 2020), indirect production via grazing (Schmale et al., 2018) and abiotic photoproduction (Li et al., 2020).  A recent study demonstrated that $CH_4$ production by cyanobacteria is linked to general cell metabolism and does not rely on the presence of methylated precursor compounds (Bižić 
[revised manuscript text omitted]

Combining this approach with new analytical techniques such as cavity enhanced absorption spectroscopy (CEAS) and non-dispersive infrared (NDIR) should continue to improve the quality of $CH_4$ and $N_2O$ flux estimates (McDermitt et al., 2011; Nemitz et al., 2018; Maher et al.,

2019). Indeed, eddy flux towers aboard ships (Thornton et al., 2020) and in coastal locations (Yang et al., 2016; Gutiérrez-Loza et al., 2019) are now being equipped with $CH_4$

instrumentation that enables the integration of $CH_4$ fluxes over large areas. There are fewer $N_2O$

flux estimates made with CEAS and NDIR and the implementation of $N_2O$ sensors on eddy flux towers remains limited. Recently, $N_2O$ emissions from Eastern Boundary Upwelling Systems were quantified using inversion modeling based on atmospheric measurements from coastal monitoring stations highlighting the potential of this approach to constrain $N_2O$ emissions from remote oceanographic regions that have significant spatial and temporal heterogeneity (Ganesan et al., 2020; Babbin et al., 2020). Inverse modeling of atmospheric measurements was also recently used to constrain $CH_4$ emissions from the East Siberian Arctic Shelf (Tohjima et al.,

2020)

[revised manuscript text omitted]
. Currently, there is no community-defined level of analytical uncertainty to characterize high quality $CH_4$ and $N_2O$ measurements. However, attaining an analytical agreement between multiple laboratories of ≤1% is considered achievable for the repeat oceanographic surveys and time-series observations (Fig. 3). For context, an analytical agreement of ≤1% would permit the ocean's response to the increasing tropospheric $CH_4$ and $N_2O$ mole fractions to be resolved on timescales of 10 and 5 years, respectively. These values are based on the changes in surface ocean $CH_4$ and $N_2O$ concentrations that are predicted to occur due to the ongoing increase in tropospheric $CH_4$ and $N_2O$ mole fractions at a seawater temperature of 20°C and a salinity of 35 g kg$^{-1}$, and assuming all sources and sinks remaining constant. In our recent marine $CH_4$ and $N_2O$ inter comparison exercise we concluded that the diversity of analytical procedures employed by the participants was a major cause of high variability between the reported concentrations, highlighting an urgent requirement for $CH_4$ and $N_2O$ SOPs (Wilson et al., 2018). Consequently, these SOPs are now being compiled, and they will be freely available via the Ocean Best Practices System.

(ii) The second is increased regularity of intercomparison exercises through the periodic distribution of consensus material, i.e. water samples in which $CH_4$ and $N_2O$ concentrations are known with high confidence, obtained by pooling analyses from several laboratories with demonstrated analytical capability. These will help the scientific community to monitor data comparability and accuracy, particularly in the case of highly elevated concentrations of $CH_4$ and $N_2O$, i.e. those exceeding atmospheric equilibrium concentrations by at least an order of magnitude.

(iii) The third activity builds on the previous initiative and calls for the production of Global Data Products for dissolved $CH_4$ and $N_2O$ measurements. To date, individual $CH_4$ and $N_2O$ measurements are represented at the global scale by the MEMENTO database which has been very successful at compiling $CH_4$ and $N_2O$ datasets and making them readily accessible to the modeling community. However, the MEMENTO database does not currently include a Global

Data Product that includes publicly accessible quality controlled dissolved $CH_4$ and $N_2O$ datasets. The international marine carbon science community has widely embraced such an approach for $fCO_2$, by submitting data to the Surface Ocean $CO_2$ Atlas (SOCAT), which was initiated in response to the need for a quality controlled, publicly available, global surface $CO_2$ dataset (e.g. Bakker et al., 2016). Due to the fewer measurements, a similar data product for marine $CH_4$ and $N_2O$ would be needed every ~5 years. We consider the production of Global Data Products for dissolved $CH_4$ and $N_2O$ to be essential for supporting future global modeling efforts and to enhance field observations.

The benefits of pursuing the three activities described above have already been clearly demonstrated for carbon system measurements in the ocean. The intercomparability and high accuracy and precision of carbon system measurements was achieved by streamlining methodological approaches, universally adopting agreed SOPs, production of reference material, and following community-driven quality control procedures (Dickson et al., 2007, Dickson et al, 2010). It is encouraging to see the marine $CH_4$ and $N_2O$ community beginning to move in a similar direction.

*Acknowledgements*: The workshop was held at the University of California Los Angeles Lake Arrowhead conference center during 28-31 October 2018 (https://web.whoi.edu/methane-workshop/). We are grateful to all the participants who made valuable scientific contributions to the workshop and we thank S. Ferrón for critical comments to the manuscript. The workshop was sponsored by the Ocean Carbon and Biogeochemistry (OCB) Project Office, which is supported by the U.S. National Science Foundation OCE (1558412) and the National Aeronautics and Space Administration (NNX17AB17G). The workshop received additional funding from the Moore Foundation and the Scientific Committee on Ocean Research (SCOR) which receives funding from the U.S. National Science Foundation (Grant OCE-1840868) and contributions by additional national SCOR committees. The Chilean COPAS $N_2O$ time-series measurements received financial support from FONDECYT (1200861). This is JISAO contribution number 2020-1080 and PMEL contribution number 5126

[revised manuscript text omitted]
 submesoscale level (see Fig. 5). The low resolution oceanographic surveys are more likely to achieve a high level of analytical accuracy compared to high resolution coastal measurements, however this is compensated for by high temporal resolution achieved by underway sampling.

[Figure]

Figure 4. Distributions and emissions of marine $CH_4$ and $N_2O$, (a) Air-sea $N_2O$ disequilibrium mapped using a Regression Forest model (adapted from Yang et al., 2020), (b) Air-sea $CH_4$

disequilibrium mapped using an Artificial Neural Network model (adapted from Weber et al.,

2019). For consistency with the original publications, the air-sea disequilibrium is shown in different units for $N_2O$ (partial pressure) and $CH_4$ (concentration). (c) A summary of global ocean $CH_4$ and $N_2O$ emissions estimated by Yang et al. (2020) and Weber et al. (2019), compared to the estimates of the IPCC 5th Annual Report (IPCC AR5) and the Global Methane

Budget (Saunois et al., 2016).

[Figure]

Figure 5. Key environmental predictors of surface ocean CH$_4$ and N$_2$O gradients. (a) Excess air- sea N$_2$O is best predicted by O$_2$ concentrations in the subsurface water-column (base of the mixed layer to a depth of 100 m) (adapted from Yang et al., 2020). (b) Excess CH$_4$ is best predicted by seafloor depth, reflecting the supply from anoxic sediments (adapted from Weber et al., 2019). The grey dots represent individual data points and the red dots with error bars represent mean ±1s.d. of binned data, using O$_2$ bins of 10 μM width and seafloor depth bins of

10 m width.